# Eliciting Reasoning in Language Models with Cognitive Tools

**Brown Ebouky**
IBM Research - Zurich
ETH Zurich
Brown.Ebouky@ibm.com

**Andrea Bartezzaghi**
IBM Research - Zurich
abt@zurich.ibm.com

**Mattia Rigotti**
IBM Research - Zurich
mrg@zurich.ibm.com

## Abstract

The recent advent of reasoning models like OpenAI's o1 was met with excited speculation by the AI community about the mechanisms underlying these capabilities in closed models, followed by a rush of replication efforts, particularly from the open source community. These speculations were largely settled by the demonstration from DeepSeek-R1 that chain-of-thought and reinforcement learning (RL) can effectively replicate reasoning on top of base LLMs. However, it remains valuable to explore alternative methods for theoretically eliciting reasoning that could help elucidate the underlying mechanisms, as well as providing additional methods that may offer complementary benefits.

Here, we build on the long-standing literature in cognitive psychology and cognitive architectures, which postulates that reasoning arises from the orchestrated, sequential execution of a set of modular, predetermined cognitive operations. Crucially, we implement this key idea within a modern agentic tool-calling framework. In particular, we endow an LLM with a small set of "cognitive tools" encapsulating specific reasoning operations, each executed by the LLM itself. Surprisingly, this simple strategy results in considerable gains in performance on standard mathematical reasoning benchmarks compared to base LLMs, for both closed and open-weight models. For instance, providing our "cognitive tools" to GPT-4.1 increases its pass@1 performance on AIME2024 from 32% to 53%, even surpassing the performance of o1-preview.

In addition to its practical implications, this demonstration contributes to the debate regarding the role of post-training methods in eliciting reasoning in LLMs versus the role of inherent capabilities acquired during pre-training, and whether post-training merely uncovers these latent abilities.

## 1   Introduction

The recent introduction of Large Language Models (LLMs) with reasoning capabilities has showcased the potential of unfolding test-time compute as chain-of-thought traces representing intermediate steps toward obtaining an answer to a query. The success of the first reasoning models like OpenAI's proprietary o1-preview demonstrated marked improvements on coding, mathematical, and general reasoning benchmarks [OpenAI, 2024], igniting enthusiasm across the AI community and a race to replicate these results in the open.

39th Conference on Neural Information Processing Systems (NeurIPS 2025).

While the release notes associated with these models confirmed that reinforcement learning (RL) played a crucial role in enhancing reasoning capabilities, the specific mechanisms remained opaque, fueling intense speculation within the open research community. Proposed hypotheses ranged from pipelines leveraging curated fine-grained reward labels [Uesato et al., 2022, Lightman et al., 2023, Ma et al., 2023, Wang et al., 2024] to self-correction and algorithmic approaches inspired by Monte Carlo Tree Search [Hosseini et al., 2024, Xie et al., 2024, Liang et al., 2024]. This debate was partially resolved when subsequent work by DeepSeek demonstrated that relatively simple post-training recipes combining "cold-start" supervised fine-tuning on curated reasoning traces with RL optimization on verifiable rewards [Lambert et al., 2025] could produce high-quality reasoning on par with the best closed models [Guo et al., 2025].

Recently, a critical reanalysis of the role of RL in eliciting reasoning in LLMs has added a new intriguing chapter to this story by pushing the narrative that the inherent capabilities of base models might be as important as RL (if not more) in enabling reasoning. In particular, Liu et al. [2025] observed that base models on which open reasoning LLMs are often built – like Qwen2.5-Base and DeepSeek-V3-Base – already spontaneously demonstrate strong reasoning capabilities and exhibit "Aha moment" self-reflection patterns that have been purported as indicative of emerging reasoning behavior. Yue et al. [2025] went a step further by showing that the reasoning traces generated by RL-fine-tuned models are already present in the base models' generated responses if sampled sufficiently. This observation prompts them to propose that the role of RL is to bias the generation toward samples with high reward, thereby harnessing the strong reasoning capabilities that are already inherent in the base model, rather than infusing new ones.

Given these results and their implications that RL is not strictly necessary for reasoning but is merely helping "uncover" reasoning from already strong base models, it is natural to ask what other strategies might be used to elicit reasoning. Exploring alternative methods could be valuable to help theoretically elucidate the mechanisms underlying reasoning in LLMs, as well as offering complementary approaches that may provide additional benefits.

Recent work by Kramer and Baumann [2024] pointed out that cognitive psychology and cognitive sciences in general are the obvious disciplines for investigating the mechanisms underlying reasoning. In particular, those authors took inspiration from the foundational cognitive architectures framework by Anderson et al. [1997], which posits that human reasoning arises from the structured execution of stereotyped *cognitive operations* that are orchestrated into sequences suited for problem-solving. Kramer and Baumann [2024] proposed a prompt engineering implementation of these ideas that they called "cognitive prompting", consisting essentially in prompts that are structured so as to enable LLMs to break problems into stages like goal clarification, decomposition, and integration. Cognitive prompting was shown to significantly enhance arithmetic and commonsense reasoning capabilities of LLMs.

We build upon this work by going one step further in realizing the cognitive architecture idea that reasoning comes about as the orchestrated execution of modular cognitive operations that can be flexibly structured depending on the context at hand. We argue that the cognitive prompting approach is missing the important element of *modularity*, i.e., an implementation of cognitive operations that are encapsulated as discrete tools rather than a predetermined monolithic prompt. Modularity has long been proposed as a principle to reduce interference between operations in neural networks (e.g. Soldal [2012]), and it has been shown to be associated with compositional generalization in neuroscience studies [Ito et al., 2022]. Taking inspiration from modern Agentic AI, we instantiate modular and compartmentalized cognitive operations in LLMs within a tool-calling architecture where each cognitive operation is implemented as a dedicated, self-contained function. But while in agentic tool-calling frameworks, tools are external functions or APIs (e.g., calculators, search engines) with predefined schemas that LLMs invoke to execute tasks outside their parametric knowledge, in the case of our *"cognitive tools"* they encapsulate reasoning operations within the LLM itself. Each cognitive tool's schema includes a prompt template that isolates a specific cognitive operation. When invoked, the LLM executes this prompt in a sandboxed context, generating a structured intermediate result that is fed back into the main reasoning loop. Unlike general tools, which interface with external systems, cognitive tools modularize the LLM's internal reasoning processes.

## 2 Related Work

**Reasoning Elicitation in LLMs** Efforts to elicit robust reasoning in LLMs have evolved through multiple phases, building on top of the foundational work by Wei et al. [2023] who showed that Chains-of-Thought (CoT) – generated responses with intermediate reasoning steps toward an answer – can be elicited through prompting. This work was then built upon resulting in more sophisticated and effective reasoning schemes [Yao et al., 2023, Besta et al., 2025]. While reinforcement learning (RL) had a fundamental role in enabling LLMs to follow instructions from human feedback [Ouyang et al., 2022], the clearer demonstration of its role in reasoning through CoTs was demonstrated by Lightman et al. [2023] thanks to the proposal of fine-grained Process Reward Models. The subsequent idea of harnessing Process Reward Models at test-time gave rise to several works, like for instance the paper by Liang et al. [2024] who used them in search-based methods inspired by Monte Carlo Tree Search (MCTS) to improve planning accuracy. Shao et al. [2024] proposed GRPO, which Guo et al. [2025] used to cement the importance of reinforcement learning as part of the LLMs post-training pipeline for CoT-based reasoning. More recently however, analyses by Liu et al. [2025] and Yue et al. [2025] suggested that base LLMs inherently possess latent reasoning capabilities, which RL post-training merely strengthens by biasing the generation toward high-reward CoT traces.

**Cognitive Architectures and Structured Reasoning** Cognitive architectures like ACT-R [Anderson et al., 1997] were based on the assumption that human reasoning emerges from the orchestrated execution of modular operations, such as goal management and procedural memory. Kramer and Baumann [2024] proposed a first prompt-engineering translation of these principles to LLMs by introducing cognitive prompting, which structures prompts into stages like decomposition and integration, significantly improving arithmetic and commonsense reasoning. However, this approach lacks explicit modularity and compartmentalization, risking interference between reasoning stages, and limiting the flexibility of how cognitive operations can be organized. Sumers et al. [2024] provided another effort to unify cognitive architectures and LLMs, by positioning LLMs as central controllers in agentic systems with modular memory components and structured action spaces, and distinguishing external actions (e.g., API calls) from internal actions (e.g., reasoning, retrieval).

**Agentic AI Frameworks and Tool-Calling Workflows** Modern agentic frameworks, such as Toolformer [Schick et al., 2023] and HuggingGPT [Shen et al., 2023], enable LLMs to interact with external tools (e.g., calculators, APIs) via structured schemas. Recent architectures like LangChain [Chase, 2023] emphasize workflow orchestration but remain agnostic to the internal structure of reasoning steps. Our work reformulates internal cognitive operations from cognitive architectures as tools within a modern tool-calling agentic framework, by encapsulating reasoning stages into modular prompt-driven tools.

## 3 Methodology: Cognitive Tools

We propose using cognitive tools to elicit the reasoning capabilities of LLMs. We identify four cognitive tools: **understand question**, **recall related**, **examine answer**, and **backtracking**. For a given question, the LLM is prompted to use tools as needed to help it solve the problem correctly by guiding its reasoning. The execution pipeline is similar to any tool-calling pipeline: the LLM is prompted to generate a reasoning trace in response to a query until a call to one of the provided tools $t$ is issued. Once a tool call is detected, we stop the generation and execute the module that encapsulates the tool $t$. In our case, each tool represents a call to an LLM (the same as the one reasoning) with the specific tool role. The output of the execution is provided back to the LLM that issued the tool call, which continues to reason about the problem until the end-of-sequence token. This procedure is related to token and budget forcing with test-time scaling, as introduced in [Muennighoff et al., 2025]. In our work we provide additional flexibility by allowing the LLM to select which tool to call and when to call it, as if the LLM were left to autonomously and flexibly implement budget forcing when deemed appropriate.

**Understand Question** The cognitive architectures literature [Anderson et al., 1997] emphasizes the importance of goal management in replicating human reasoning, which operates by breaking down a problem at hand to identify its key components. We implement this process into what we call the "understand question" tool. The role of this cognitive tool is to prompt the LLM to break down

**A) Cognitive Prompting**

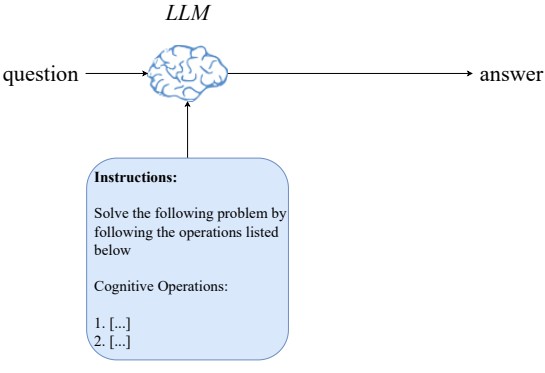

**B) Cognitive Tools**

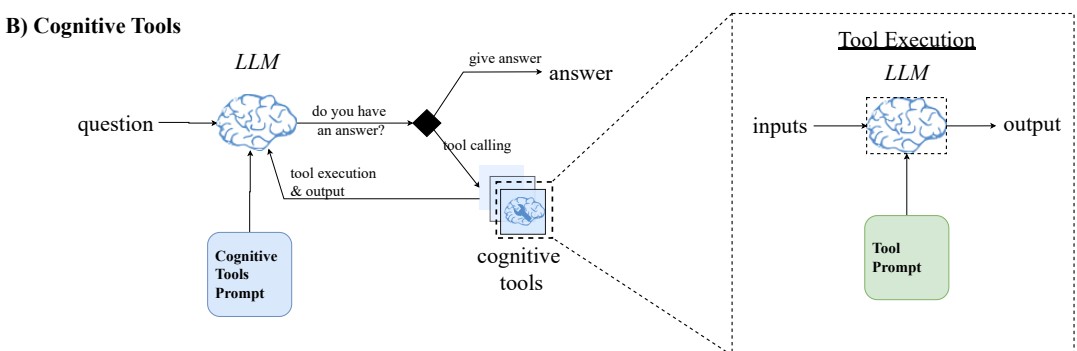

Figure 1: Overview of our Cognitive Tools pipeline vs Cognitive Prompting

the problem by identifying the main concepts, extracting relevant information in the question, and highlighting meaningful properties, theorems, and techniques that might be helpful in solving the problem.

**Recall Related**   This tool is inspired by the work Yasunaga et al. [2024], which introduces a prompting technique consisting of asking a model to recall previous knowledge to guide its own reasoning. In our case, for a given question, the tool provides relevant related knowledge from similar questions, which it knows how to answer, together with the corresponding answer. The objective is then to guide the LLM through those examples towards a way it can follow to solve the question at hand.

**Examine Answer**   The role of this cognitive tool is to examine the current trace of reasoning of the LLM when trying to find the answer to a question. In other words, it implements a form of 'self-reflection', an operation which has been demonstrated to be effective for reasoning [Shinn et al., 2023]. In practice, this cognitive tool checks the current reasoning trace for possible flaws, wrong assumptions, miscalculations, or constraints which are not taken into account. Thus, it helps the LLM to reconsider its reasoning and correct any oversights.

**Backtracking**   When one is faced with an incorrect solution to a problem or realizes that the train-of-thoughts is flawed, the next action is to backtrack to a previously correct step and explore alternatives, an idea related to Monte Carlo Tree search (see e.g. Liang et al. [2024]). This defines the idea behind this tool, which is to enable *exploration* of more promising reasoning paths. When the LLM selects a cognitive tool, the tool prompts the LLM to consider the current reasoning trace, summarizing it and breaking it down into steps. The LLM then proceeds to evaluate which step in the reasoning process is incorrect and follows up by providing alternative approaches or directions for solving the problem.

The cognitive tools are provided to the LLM together with a *cognitive tools prompt*, i.e. a system prompt that instructs the LLM on how to proceed with the reasoning process, reading as follows:

---

**Cognitive Tools Prompt**

You are an expert assistant who solves problems thoughtfully and effectively. You have access to a list of tools — these are Python-based functions that you can call to help you reason through or solve the problem more efficiently.

You are encouraged to use tools when they make the task easier, clearer or more robust — especially for complex, elaborated or ambiguous questions.
Use your best judgment to decide when to call tools.

You may call tools at any point in your reasoning process. Only use the tools listed below. If you choose to use a tool, describe your reasoning and clearly call it using their name.
You can solve problems however you find most appropriate.
When you are ready to provide the final answer to the problem or the question always follow the syntax: 'ANSWER: answer'.

You only have access to these tools, do not use any others:

{{cognitive_tools_signature}}

Here are the rules you should always follow to solve your task:

1. **Call a tool when needed.** If you call a tool, only use the available ones and use its full name to do so.
2. ONLY USE Python to call an available tool and not for something else.
3. Don't give up! You're in charge of solving the problem.
4. Do not give an answer without reasoning about it.
5. **Never hallucinate results.** Wait for tool responses before continuing.
6. **Only write your final answer** after you are confident, and always in the form: 'ANSWER: your final answer here'

If the question is already clear, you may skip the 'understand_question' step when the corresponding tool is available. But when unsure, it's good practice to use it.

Now Begin! If you solve the task correctly, you will receive a reward of $1,000,000.

---

The placeholder $\{\{cognitive\_tools\_signature\}\}$ is replaced with the tools which the LLM can use to help its reasoning. In order to take advantage of the capability of LLMs to generate code in addition to using the cognitive tools, we enable the LLM to generate code as an additional modular reasoning tool.

In the Appendix, we provide more details on how each tool is individually implemented. Figure 1 provides an illustration of our cognitive tools pipeline compared to cognitive prompting. The main LLM selects the tool to be executed; the selected tools are executed independently from the main LLM, but using the same instance, in a modular approach; the execution output is then fed back to the main LLM, which continues working on the response until the final answer is generated. We provide pseudo-code of our cognitive tools pipeline in the Appendix.

## 4  Experiments

**Datasets**   Following established evaluation practices in the reasoning literature [Hendrycks et al., 2021], in this work we investigate the elicitation of reasoning using cognitive tools on math-oriented benchmarks. We focus our experiments on math benchmarks because of how reasoning is central in solving math problems. Specifically, we consider the following datasets:

- **AIME 2024** [MAA, 2024] is a dataset that contains 30 samples which are problems used in the 2024 American Invitational Mathematics Examination (AIME) held from January 31 - February 1, 2024. It is a prestigious high school mathematics competition known for its challenging mathematical problems on arithmetic, algebra, counting, geometry, number theory, probability, and other secondary school math topics.
- **MATH 500** [Hendrycks et al., 2021] contains 500 math problems across subjects similar to those in AIME2024 [MAA, 2024] and with different difficulty level.
- **AMC** [Li et al., 2024] is a curated collection of 83 problems from the AMC competitions of 2022 and 2023, which provides challenging math problems.
- **Smolagents Benchmark-v1** [Huggingface, 2024] is composed of questions about different tasks, such as math or question answering from HuggingFace. Specifically, we consider the math task (50 samples) and refer to it as *Smolbenchmark* in the rest of the paper.

**Models** We use the open-weight models Qwen2.5-(7B, 32B) Instruct [Qwen Team, 2024], Llama3.1-8B Instruct, and Llama3.3-70B Instruct [AI@Meta, 2024]. We also experiment with closed models GPT-4.1 and o1-preview.

**Evaluation and Baselines** In all experiments, we report the model's accuracy in providing the correct answer on the first try (pass@1). For AIME 2024 [MAA, 2024] and AMC [Li et al., 2024], the answer from the model is compared to the ground truth via parsing. Regarding MATH500 [Hendrycks et al., 2021], which includes more elaborated answers that are not just numerical (e.g., complex expressions), we use an LLM-as-a-judge approach to establish the veracity of the answers [Zheng et al., 2023]. Specifically, we use GPT-4.1 as a judge and report the accuracy of the model in answering the questions (see the prompt used for the judge LLM in the Appendix).

Our baseline represents the accuracy out-of-the-box of a model which is prompted to solve the question. We compare these results to those from our cognitive tools framework and we additionally provide results when cognitive prompting is used.

## 5 Results

### 5.1 Reasoning with Cognitive Tools

Firstly, we are interested to understand how useful our cognitive tools are in helping an LLM to solve a problem. We introduced in Section 3 cognitive operations encapsulated in the four cognitive tools: 'understand question', 'recall related', 'examine answer' and 'backtracking'. In the following, we report the effect of adding each tool individually to an LLM to understand for each LLM in our test suite which tools are more helpful, since different LLMs reason differently. Table 1 first shows the accuracy achieved by the LLMs on Smolbenchmark (baseline). It then shows the accuracy of the LLMs endowed with each tool individually. We observe that each cognitive tool allows the LLMs to outperform the baseline, with even a $+26.7\%$ jump on Llama3.3-70B Instruct using the 'understand question' tool. The impact of each tool varies between the models, with different tools providing the best improvements. Despite that, we clearly see that our cognitive tools generally help in the reasoning of the LLM towards the correct solution of a problem.

### 5.2 Cognitive Tools vs Cognitive Prompting

Our work builds on the concept of cognitive prompting presented by Kramer and Baumann [2024], complementing it with the insight from cognitive psychology and theoretical neuroscience that modularity might be important, as it is a fundamental component of cognitive architecture [Anderson et al., 1997] and has been shown to enable compositional generalization [Ito et al., 2022]. In addition, a modular approach has multiple benefits from a prompt-engineering perspective compared to a monolithic prompting approach. First, modularity helps the LLM to focus on implementing the specific cognitive operation at hand, in isolation from the rest of the context window that has been provided so far. In other words, it reduces interference from all the information that has been provided to the LLM throughout the reasoning process, as the LLM is only provided with specific prompts and the limited information from the context windows corresponding to the inputs of the cognitive tool specified in its schema. Second, our modular approach encourages flexibility: we do not enforce

| Tools | Qwen2.5-7B | Qwen2.5-32B | Llama3.1-8B | Llama3.3-70B |
|---|---|---|---|---|
| baseline | $75.8 \pm 1.1$ | $79.6 \pm 1.4$ | $48.7 \pm 1.8$ | $52.8 \pm 1.2$ |
| understand question | $78.6 \pm 0.7$ | $82.5 \pm 0.8$ | $\mathbf{59.4} \pm 0.9$ | $\mathbf{79.5} \pm 0.8$ |
| recall related | $76.1 \pm 0.8$ | $\mathbf{84.2} \pm 0.8$ | $53.2 \pm 1.5$ | $75.1 \pm 0.8$ |
| examine answer | $77.8 \pm 0.8$ | $84.0 \pm 0.6$ | $50.9 \pm 1.3$ | $74.9 \pm 0.7$ |
| backtracking | $\mathbf{80.5} \pm 0.5$ | $82.9 \pm 0.8$ | $57.2 \pm 1.6$ | $78.2 \pm 1.0$ |

Table 1: Accuracy of the 'Instruct' version of the listed model on the Smolbenchmark dataset. 'Baseline' indicates the performance of the plain model. The subsequent rows indicate the performance of the models endowed with each specific cognitive tool. Tools generally provide a boost over the baseline with different tools achieving the highest performance for different models. The values in the table are average pass@1 accuracy over 16 repetitions, and uncertainty intervals represent standard error. A Welch's t-test confirms that all differences between baseline and the best individual tool are statistically significant with $p < 0.05$

the model to use a predefined order of tool calls or strategy to solve the query, but instead we let it figure out the best way to answer the question. This contrasts with cognitive prompting [Kramer and Baumann, 2024], which directly provides the LLM with the order of steps it needs to follow to solve a given problem.

To concretely demonstrate these presumed benefits of modular cognitive tools over monolithic prompting approaches like Kramer and Baumann [2024] we compare the performance of cognitive prompting with our cognitive tools on Smolbenchmark, and provide the results in Table 2. We notice that, while cognitive prompting reliably matches or outperforms the baseline model, our modular cognitive tools consistently surpass cognitive prompting in performance. In particular, we obtain performance increases ranging from $+4.2\%$ on Qwen2.5-7B-Instruct to $+27.2\%$ on Llama3.3-70B over the baseline. These results confirm the effectiveness of our cognitive tools approach in eliciting robust reasoning in LLMs.

| Tools | Qwen2.5-7B | Qwen2.5-32B | Llama3.1-8B | Llama3.3-70B |
|---|---|---|---|---|
| baseline | 75.8 | 79.6 | 48.9 | 52.8 |
| cognitive prompting | 74.0 | 82.0 | 47.1 | 66.0 |
| cognitive tools | **80.0** | **88.0** | **60.0** | **80.0** |

Table 2: Comparison between baseline (regular LLM without tools), cognitive prompting [Kramer and Baumann, 2024] and cognitive tools on the Smolbenchmark dataset (all model used are the `Instruct` instruction fine-tuned version).

### 5.3 Main Results

Figure 2 provides the main results demonstrating the effectiveness of cognitive tools to solve challenging math problems. In the figure we show the performance of our cognitive tools when added to the baseline model and report in Table 3 more detailed results. For this benchmark, models have all the cognitive tools that we examined in Table 1 at their disposal. We report the accuracy of both the baseline and the model with cognitive tools on AIME 2024 [MAA, 2024], MATH500 [Hendrycks et al., 2021], AMC [Li et al., 2024]. We also include the results of Chain-of-Thought (CoT) [Wei et al., 2023] and code-equipped baseline model. For the latter, we include a prompt allowing the model to write code, which is then executed and the output given back to the model to continue its reasoning. We observe that on AIME 2024, despite it being a very difficult dataset, our cognitive tools are able to improve significantly over the baseline. The availability of cognitive tools consistently improves performance across all models and benchmarks. This further validates that our cognitive tools transfer well across reasoning benchmarks like math problems. Furthermore, while equipping the model with coding capabilities is beneficial, our cognitive tools continue to play a crucial role in enhancing reasoning capabilities.

| Model | AIME 2024 | MATH500 | AMC | Avg |
|---|---|---|---|---|
| Qwen2.5-7B Instruct | 12.5 ± 0.7 | 71.7 ± 1.3 | 43.9 ± 1.3 | 42.7 |
| Qwen2.5-7B Instruct + cot | 12.5 ± 0.1 | 71.8 ± 0.4 | 41.5 ± 0.7 | 41.9 |
| Qwen2.5-7B Instruct + code | 12.1 ± 0.6 | 73.1 ± 0.5 | 42.9 ± 0.7 | 42.7 |
| Qwen2.5-7B Instruct + **cognitive tools** | **14.6** ± 1.8 | **73.7** ± 0.5 | **47.0** ± 0.5 | **45.1** |
| Qwen2.5-32B Instruct | 17.2 ± 1.2 | 74.1 ± 0.7 | 52.6 ± 0.8 | 48.0 |
| Qwen2.5-32B Instruct + cot | 15.4 ± 0.6 | 79.2 ± 0.3 | 50.4 ± 0.6 | 48.3 |
| Qwen2.5-32B Instruct + code | 19.6 ± 0.6 | 80.6 ± 0.3 | 57.7 ± 0.8 | 50.2 |
| Qwen2.5-32B Instruct + **cognitive tools** | **32.1** ± 1.9 | **81.8** ± 0.6 | **62.7** ± 1.2 | **58.9** |
| Llama3.1-8B Instruct | 5.8 ± 1.0 | 43.2 ± 0.5 | 20.3 ± 0.8 | 23.1 |
| Llama3.1-8B Instruct + cot | 7.9 ± 1.6 | 53.3 ± 0.5 | 24.5 ± 1.3 | 28.6 |
| Llama3.1-8B Instruct + code | 5.8 ± 1.3 | 51.6 ± 0.7 | 26.7 ± 1.0 | 28.0 |
| Llama3.1-8B Instruct + **cognitive tools** | **8.8** ± 1.7 | **50.7** ± 1.0 | **28.0** ± 1.2 | **29.2** |
| Llama3.3-70B Instruct | 13.1 ± 1.0 | 57.0 ± 0.5 | 33.0 ± 0.9 | 34.4 |
| Llama3.3-70B Instruct + cot | 18.1 ± 1.0 | 70.7 ± 0.5 | 40.6 ± 0.8 | 43.1 |
| Llama3.3-70B Instruct + code | 19.0 ± 0.7 | 71.6 ± 0.3 | 45.2 ± 1.2 | 45.3 |
| Llama3.3-70B Instruct + **cognitive tools** | **29.8** ± 1.2 | **74.7** ± 0.5 | **51.0** ± 0.5 | **51.8** |

Table 3: Evaluation of our cognitive tools pipeline for different base LLMs on the math benchmarks detailed in Section 3. The availability of cognitive tools enable LLMs to display robust reasoning which consistently results in significant improvement in pass@1 accuracy (the table shows averages over multiple runs ($> 8$) and uncertainty intervals representing standard error).

## 5.4 How close are we to a reasoning model (RL trained)?

One hypothesis of our work is that it is possible to find alternatives to elicit reasoning capabilities of LLMs other than RL. We proposed that modular cognitive tools could be a possible mechanism of achieving this and demonstrated the viability of this approach on open-weight models to tackle challenging math reasoning tasks.

An intriguing outstanding question is how well this type of reasoning would compare against reasoning models, in particular on closed models which provided the first demonstration of reasoning capabilities. To take a step towards answering this question, we add our cognitive tools to GPT-4.1 and evaluate the accuracy of the augmented model with respect to its baseline as well as o1-preview reasoning, the first reported reasoning model trained with RL. We perform this evaluation on AIME 2024 and Table 4 shows the results of this experiment. We observe that GPT-4.1 with cognitive tools significantly outperforms the baseline, achieving performance that surpasses o1-preview, all without any additional training, solely through the reasoning enhancements provided by our cognitive tools.

| Model | AIME 2024 |
|---|---|
| o1-preview | **44.6** |
| GPT-4.1 | 32.0 |
| GPT-4.1 + **cognitive prompting** | 42.0 |
| GPT-4.1 + **cognitive tools** | **53.0** |

Table 4: GPT-4.1 vs o1-preview on AIME 2024.

## 6 Discussion and Conclusion

Our work proposes a method for eliciting reasoning in LLMs that is based on the idea of modular *cognitive tools*, implemented as isolated, prompt-driven operations within an agentic tool-calling framework. We demonstrate that cognitive tools substantially improve the reasoning performance of

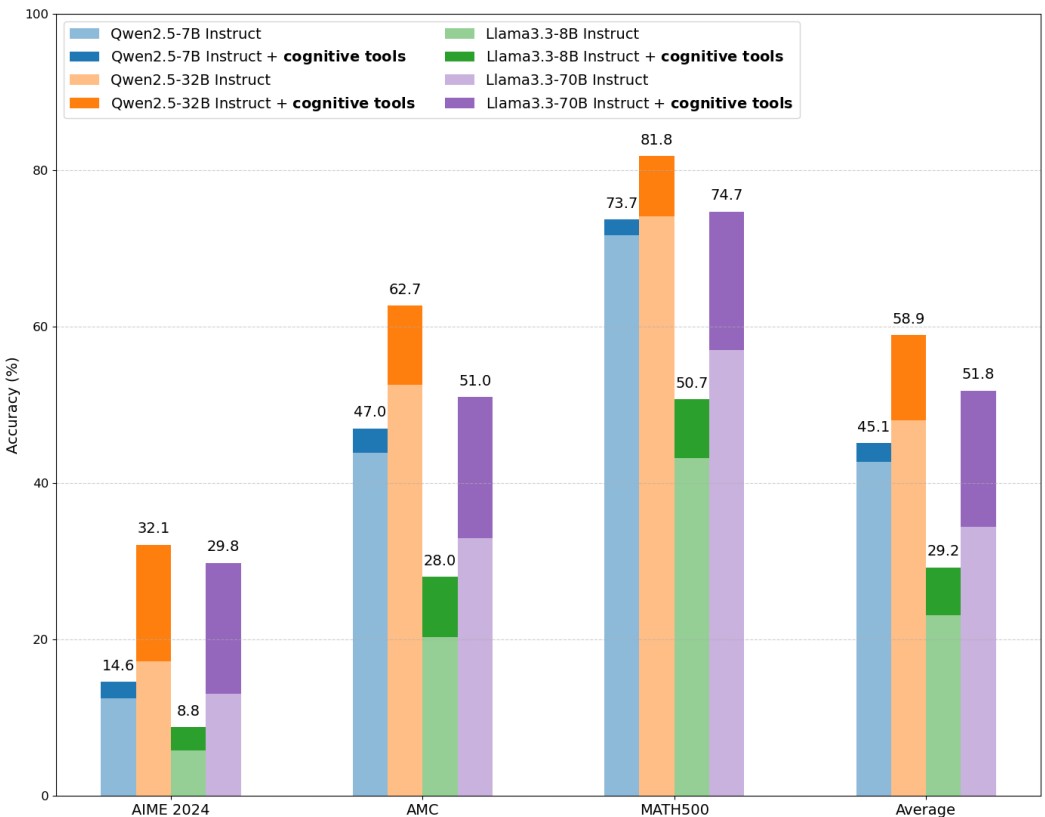

Figure 2: Bar plot displaying the gains in terms of pass@1 accuracy provided by our cognitive tools for different models over the math reasoning benchmarks AIME2024, AMC and MATH500. The lighter shades represent the baseline performance, the values refer to the performance of the LLMs endowed with cognitive tools, and the darker shades represent the relative gain. The plotted valued are reported in Table 3.

large language models. By compartmentalizing cognitive operations such as query understanding, recalling, re-examination, and backtracking, our approach reduces interference between reasoning steps, which addresses a key limitation of flat prompting and monolithic chain-of-thought methods. The observed gains in mathematical reasoning benchmarks highlight the practical value of this modular design.

Our findings also contribute to the ongoing debate about the origins of reasoning in LLMs. The effectiveness of base models endowed with cognitive tools supports the hypothesis that pre-training instills latent reasoning capabilities, which can be surfaced through structured modular workflows, rather than being capabilities that are instilled via post-training, for instance via reinforcement learning. This indicates that modular prompting could be a more interpretable and possibly more efficient alternative or complement to reinforcement fine-tuning.

From an agentic AI perspective, our approach bridges the gap between traditional tool-calling – focused on external APIs and functions – and the need for explicit, modular internal reasoning. This not only aligns with insights from cognitive science and neuroscience regarding the benefits of modularity and compositionality, but, since each reasoning step can be associated with a particular cognitive tool, also enhances transparency and explainability in AI agents, which is crucial for real-world applications demanding interpretable decision-making.

## Broader Impact and Limitations

Our framework for modular cognitive tools carries significant implications for developing transparent and reliable AI systems. By structuring reasoning into discrete, interpretable steps, it enhances accountability in contexts where understanding AI decision-making is crucial, particularly in high-stakes domains such as healthcare, education, and legal analysis. The method's compatibility with base models also democratizes access to advanced reasoning capabilities, reducing reliance on resource-intensive reinforcement learning pipelines. However, while promising, our method currently relies on manually defined cognitive tools and has been primarily evaluated on mathematical reasoning tasks tailored to the specific models tested. In other words, the prompts that implement our cognitive tool might not work as well on model families other than those that we tested without additional prompt-engineering effort. Future work should explore the automated discovery of cognitive operations, integrate with reinforcement learning, and expand applications to domains beyond arithmetic reasoning.

## Acknowledgments

This work is partly funded by the European Union's Horizon Europe research and innovation program under grant agreements No. 101070408 (SustainML) and was supported by the Swiss State Secretariat for Education, Research and Innovation (SERI) under contract number 22.00295.

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

# Appendix

## A  Experiments Details

In this section, we provide more details on the experiments. Specifically, we include the hyper-parameters used by the models, the prompts used for the baseline, for the cognitive prompting experiments and for each cognitive tool implementation.

### A.1  Model Inference Hyper-parameters

We use the default hyper-parameters provided in the model configurations for all the models that we considered in our experiments. For instance, for Qwen2.5-(7B,32B)-Instruct we use a temperature of $0.7$, top-p of $0.8$ and top-k of $20$. As for Llama-3.1-(8, 70)B-Instruct, the temperature is of $0.6$, and top-p of $0.9$.

### A.2  Baseline

We establish our baseline on Qwen2.5-(7B, 32B) Instruct, Llama3.1-8B Instruct, Llama3.3-70B Instruct and GPT-4.1 models by prompting the LLM with the question we want to have an answer for. We only append the sentence: *"Solve the math problem: "* to each question and we do not change the system prompt of the model. The final prompt to the LLM looks like:

> **Solve the math problem:** 'Rick is thinking of a positive factor of $14$ and Steve is thinking of a positive factor of $42$. If Rick and Steve are thinking of the same number, how many possible numbers could they be thinking of?'

### A.3  Cognitive Prompting

For the cognitive prompting strategy, we use the prompt released in Kramer and Baumann [2024], which is as follows:

---

**Cognitive Prompting (prompt)**

Solve the following math problem by following each step of cognitive operations from the list below. For each step, provide your reasoning and calculations before moving on to the next step.

Cognitive Operations:
1. Goal Clarification: Restate the problem in your own words.
2. Decomposition: List the given information.
3. Filtering: Identify what you need to find.
4. Reorganization: Assign variables to the unknowns.
5. Pattern Recognition: define each variable clearly.
6. Abstraction: Set up equations based on the problem.
7. Generalization: Solve the equations step by step.
8. Integration: Verify your solution with the given information.

Your Response: Please start with "Restate the problem in your own words" and proceed through each cognitive operation step by step, providing detailed reasoning and calculations for each. Give the final answer using the format: 'ANSWER: answer'.

---

### A.4  Pseudo-code of cognitive tools pipeline

We report in Algorithm 1 pseudo-code illustrating how tools interact with the main LLM loop in our cognitive tools pipeline.

**Algorithm 1** LLM-Orchestrated Reasoning with Cognitive Tools
___
1: Initialize context ← {question: *question*, history: [ ]}
2: **while** True **do**
3:     response ← LLM(prompt = "Cognitive Tools Prompt", context)
4:     **if** response["action"] = "answer" **then**
5:         **return** response["answer"]
6:     **else if** response["action"] = "call_tool" **then**
7:         tool_input ← response["tool_input"]
8:         tool_name ← response["tool_name"]
9:         tool_output ← LLM(prompt = "Tool Prompt", inputs = tool_input)
10:         context["history"].append({tool_call : tool_input, tool_output : tool_output})
11:     **end if**
12: **end while**
___

## A.5   Cogntive Tool Prompts

As explained in the main text, the cognitive tools that we introduce are implemented in a modular fashion. Each cognitive tool is implemented as a call to an LLM (same as the original one) but with a specific prompt tailored to the specifics of the tool. Below we present the prompt used for each cognitive tool:

---

**Understand Question Prompt**

You are a mathematical reasoning assistant designed to analyze and break down complex mathematical problems into structured steps to help the system that actually solves problems. Your goal is to:

1. Identify the core mathematical concepts involved (e.g., algebra, calculus, linear algebra).

2. Extract and categorize relevant symbols, variables, and functions.

3. Rephrase the problem into a step-by-step sequence that makes solving easier.

4. Highlight any known theorems or techniques that might be useful in solving the problem.

5. DO NOT provide any answer to the question, only provide instructions which will guide the upstream system."

---

**Recall Related Prompt**

You are a retrieval assistant whose purpose is to help solve new mathematical problems by providing solved examples of analogous problems.

Given a new math problem, your task is to:
1. Identify 2 or 3 **similar problems** from your knowledge or training set that require **comparable mathematical concepts or reasoning steps**.
2. For each similar problem:
- Provide the **full problem statement**.
- Provide a **complete step-by-step solution**, including relevant formulas, simplifications, or code.
- Highlight the **final answer**, preferably using LaTeX formatting (e.g., '$\boxed{42}$').

Do **not** solve the current problem. Instead, present only useful analogous examples that could help someone reason through it.

Output Format:

Analogous Example 1:
Q: [Similar Problem 1]
A: [Step-by-step solution...]

---

Final Answer: `...`

Analogous Example 2:
Q: [Similar Problem 2]
A: [Step-by-step solution...]
Final Answer: `...`

Analogous Example 3:
Q: [Similar Problem 3]
A: [Step-by-step solution...]
Final Answer: `...`

Some important notes to keep in mind.

- Select examples with strong structural or conceptual similarity, not just keyword overlap.
- Variation in surface details (numbers, variable names) is acceptable as long as the mathematical logic aligns.

---

## Examine Answer Prompt

You are an expert mathematical assistant tasked with **verifying and improving** solutions to complex mathematical problems. Your role is **not to solve the problem** but to critically analyze the provided solution for correctness, clarity, and completeness. You will be given a problem/question and the current reasoning that has been produced so far.

### **Your Task:**

Follow a structured **verification process**:

### **1. Understanding the Problem**
- Ensure the proposed solution correctly interprets the given problem.
- Identify the core mathematical concepts involved (e.g., algebra, calculus, number theory).
- Extract and categorize relevant symbols, variables, and functions.
- Identify any implicit assumptions or missing constraints.

### **2. Verifying the Given Solution**
- Clearly state what is the current answer of the problem.
- Break the provided solution down into distinct logical steps.
- Check for **logical consistency**, **mathematical correctness**, and **proper justification**.
- Identify any **miscalculations, incorrect assumptions, or unjustified leaps** in reasoning.
- Analyze the **edge cases** or conditions where the solution may fail.
- Evaluate whether all necessary steps and justifications are present.

#### **2.a) Testing and Validation (Problem-Derived Checks)**
- Examine the original problem statement and extract any **constraints, conditions, identities, or testable properties** that a correct answer must satisfy.
- Derive **test cases or evaluation criteria** based on those constraints.

**If the proposed solution is a numerical answer:**
- Plug the number into the original equation(s), inequality, or scenario to verify it satisfies all conditions.
- Check whether it meets qualitative criteria (e.g., smallest, largest, integer, range bounds).

**If the proposed solution is an expression or formula:**
- **Symbolically substitute** the expression into the original problem statement or equations, and confirm that it satisfies all requirements.
- Simplify or manipulate the expression to check **equivalence**, **domain correctness**, and

**edge cases**.
- Where applicable, test the expression against representative sample inputs derived from the problem.

**For both cases:**
- Clearly describe each test performed and the outcome.
- State whether the provided answer (number or expression) **passes all derived problem-based tests**.

### **3. Suggesting Improvements**
- If an error is found, explain **precisely what is wrong** and **why**.
- Suggest possible fixes or improvements **without directly solving the problem**.
- Propose alternative methods to solve the problem where relevant (e.g., algebraic vs. numerical, direct proof vs. counterexample).

### **4. Providing a Judgment**
- Clearly state whether the proposed solution is **correct or incorrect**.
- Justify your judgment with a concise explanation.
- If incorrect, **recommend corrections** without providing a direct answer.

### **Guidelines to Follow:**
- DO NOT provide the actual answer to the problem.
- Focus only on verifying and critiquing the given solution.
- Be rigorous in checking correctness but also constructive in suggesting improvements.
- Explicitly say whether the answer is correct or incorrect

Now, **critically analyze the solution**, highlight any mistakes, and suggest improvements where necessary."""

---

**Backtracking Prompt**

You are a careful problem-solving assistant with the ability to backtrack from flawed logic.

You will be given a math or logic problem and a reasoning trace. Your task is to:
1. Analyze the reasoning and summarize it into different steps.
2. Identify where the first error, bad assumption, or confusion occurs (if any).
3. Propose how to revise the approach from that point onward, using the steps that you have defined.
4. If the entire approach was invalid, suggest a better strategy from scratch.

Use the following format for your response:

**Identified Issues:**
- Step X: Explain what is incorrect or suboptimal.
- (Repeat for any additional steps if needed.)

**Backtrack Point:**
- Indicate the step where reasoning was still valid and you can continue from.

**Revised Strategy (from backtrack point or new):**
- Present a step-by-step strategy to solve the problem correctly from this point.
—

Be precise and critical. Avoid vague judgments. Always backtrack to the most recent correct step, unless no step is valid. """

We also provide below the prompt used for the "code tool", which is called whenever the LLM attempts to generate code during the reasoning process:

> **Use Code Prompt**
>
> You are a Python coding assistant designed to generate correct and efficient code to solve a given problem or question.
>
> You will receive:
> - A **problem description** that outlines the task to solve.
> - Optionally, **chain-of-thought (CoT) reasoning** which may contain errors.
> - Optionally, a **previous attempt at code** and/or **error messages** if earlier attempts failed.
>
> Your tasks:
>
> 1. **Analyze** the problem and any provided reasoning or code.
> 2. If the reasoning or code contains **mistakes**, **ignore or fix them** as appropriate.
> 3. Generate a **correct and clean Python solution** to the original problem.
> 4. If provided with an error message, **identify the cause** and **refine the code** accordingly.
> 5. Your code must be:
> - **Correct**
> - **Efficient**
> - **Well-structured** and **readable**
> 6. ALWAYS follow this format:
>
> Thought: your thinking process on how you want to solve the problem with code which can be helped by the previous reasoning or from scratch
> Code:
> '''python <your code here> ''' 7. Ensure the code **prints the final result** using 'print()'. The result must be printed explicitly.
>
> **Important rules:**
>
> - Think first before you give out the code
> - If necessary, re-derive the correct logic yourself.
> - Prioritize correctness, even if it means deviating from flawed prior steps.
> - ALWAYS explicitly PRINT the final result in the code with 'print()'
>
> Now generate the code to solve the problem.

# B  Additional Analysis

## B.1  Ablation on motivational phrases

Our cognitive tools pipeline is made available to an LLM through *cognitive tools prompt* defined in section 3. This prompt includes motivational phrases which can encourage the LLM to solve the task at hand. For instance, we have at the end of the prompt the sentence "*Now Begin! If you solve the task correctly, you will receive a reward of $1,000,000*". In this section, we evaluate the impact of such motivational points on the performance of our pipeline. To do so, we run experiments on AIME2024 where we remove from the cognitive tools prompt any motivational phrases and leave strict instructions related to the task and the pipeline. In table 5, we show the results on Llama and Qwen models, comparing the baseline, the original cognitive tools prompt and the one without motivational cues. we observe that removing the motivational cues does not negatively impact our method. Interestingly, if anything we actually see a modest improvement in average performance across models when removing the motivational cues (although that comes at the cost of increased variability, as indicated by higher standard errors).

## B.2  Statistics on tool usage

In this section, we provide an analysis of tool calls. The first observation that this analysis reveals is that the statistics of cognitive tool calls depend considerably on the task. For instance, the "understand

| Mode | Qwen2.5-7B | Qwen2.5-32B | Llama3.1-8B | Llama3.3-70B |
|---|---|---|---|---|
| baseline | $12.5 \pm 0.7$ | $17.2 \pm 1.2$ | $5.8 \pm 1.0$ | $13.1 \pm 1.0$ |
| no motivational cues | $16.7 \pm 3.3$ | $32.1 \pm 2.9$ | $9.8 \pm 3.7$ | $36.7 \pm 3.5$ |
| original | $14.6 \pm 1.8$ | $32.1 \pm 1.9$ | $8.8 \pm 1.7$ | $29.8 \pm 1.2$ |

Table 5: Accuracy of the "Instruct" version of the listed model on the AIME2024 dataset. "baseline" indicates the performance of the plain model, "no motivational cues" indicates the performance of cognitive tools pipeline without motivational cues, and "original" shows results of the original cognitive tools pipeline.

question" tool is called quite frequently for more difficult benchmarks like AIME2024 (in which across all models it is called in 80% of the samples) with respect to the easier Smolbenchmark (in which it is called 14% of the samples), consistent with the idea that harder tasks might require a deeper reflection and initial planning. We saw a similar pattern for "examine answer" (called 60% of the time on AIME2024 and 20% of the time on Smolbenchmark), while this pattern was almost reversed for the "use code" tool (called 77% of the time on AIME2024 and 80% of the time on Smolbenchmark). Table 6 gives more details on the frequency of the tool use on the different datasets.

| Dataset | understand | examine | use code | backtrack | recall |
|---|---|---|---|---|---|
| AIME2024 | 80.7 | 61.6 | 77.0 | 55.4 | 54.3 |
| MATH500 | 58.3 | 43.0 | 72.9 | 22.4 | 22.1 |
| AMC | 74.9 | 52.7 | 73.0 | 41.5 | 40.9 |
| Smolbenchmark | 14.3 | 19.8 | 79.8 | 8.25 | 8.25 |

Table 6: Frequency (%) of tool use across all model families on the different datasets: AIME2024, MATH500, AMC, Smolbenchmark. The "understand", "examine", "backtrack" and "recall" columns refer to "understand question", "examine answer", "backtracking" and "recall related" tools, respectively.

We also saw differences across model families. For instance, Llama models called the "use code" tool more frequently on AIME2024 than Smolbenchmark (80% of the time versus 74% of the time), while Qwen models exhibited the opposite pattern, calling the "use code" tool only 58% of the time on AIME2024 but 84% of the time on Smolbenchmark. On average, with respect to the relative use of "backtracking" and "understand question", our analysis confirms the intuition that the latter is more useful as it is being called 42% of the time across benchmarks and models, while "backtracking" is only called 20% of the time. The tool "examine answer" lies in between, with an average call frequency of 35%. We provide in Table 7 more details on the frequencies of tool use relatively to the model families and datasets.

| Model Family | Dataset | understand | examine | use code | backtrack | recall |
|---|---|---|---|---|---|---|
| Llama | AIME2024 | 99.3 | 74.4 | 80.1 | 64.1 | 61.4 |
| | MATH500 | 96.5 | 72.1 | 67.9 | 36.4 | 34.3 |
| | AMC | 97.9 | 69.5 | 74.1 | 50.8 | 48.6 |
| | Smolbenchmark | 25.8 | 36.9 | 74.3 | 15.6 | 15.2 |
| Qwen | AIME2024 | 16.3 | 0.75 | 58.3 | 0.71 | 5.75 |
| | MATH500 | 4.88 | 2.07 | 80.0 | 0.18 | 23.1 |
| | AMC | 12.1 | 0.87 | 67.0 | 0.57 | 6.28 |
| | Smolbenchmark | 2.18 | 1.37 | 84.04 | 0.21 | 0.71 |

Table 7: Details on the frequency (%) of tool use across all Llama and Qwen model families on the different datasets: AIME2024, MATH500, AMC, Smolbenchmark. The "understand", "examine", "backtrack" and "recall" columns refer to "understand question", "examine answer", "backtracking" and "recall related" tools, respectively.

The tools "backtracking" and "recall related" are the least used (both around 20% of the time), but, interestingly, they tend to be called in conjunction as they display the highest correlation between tool calls (a Pearson correlation of 0.18 on Smolbenchmark). A possible explanation for this correlation is that the LLMs are implementing a sort of "recovery workflow", where they first recall multiple relevant pieces of information, try to follow them through in sequence, and backtrack upon encountering failure, to then pursue the next one. For more details, we provide in Table 8 the Pearson correlation of the tool calls, showing how often one tool is called in combination with another one.

|            | understand | examine | backtrack | recall |
|------------|------------|---------|-----------|--------|
| understand | 1.000      | 0.005   | 0.136     | 0.125  |
| examine    | 0.005      | 1.000   | 0.009     | 0.003  |
| backtrack  | 0.136      | 0.009   | 1.000     | 0.181  |
| recall     | 0.125      | 0.003   | 0.181     | 1.00   |

Table 8: Pearson correlation between cognitive tools, defining the co-occurrence of their use. The calculation is done over on Smolbenchmark and across model families (Llama, Qwen). The "understand", "examine", "backtrack" and "recall" columns refer to "understand question", "examine answer", "backtracking" and "recall related" tools, respectively.

### B.3 Computational Overhead

We provide in this section the computational cost of our cognitive tools pipeline through the average number of output tokens. Indeed, for our GPT-4.1 experiments, we calculated the average token counts per question when running our cognitive tools on AIME2024 and we obtained 4,200 output tokens per question compared to 2,000 output tokens on the baseline. This shows that cognitive tools incur a cost in terms of output tokens that is more than twice the non-reasoning baseline, which is consistent with the higher cost of reasoning models, and emphasizes the known trade-offs between accuracy and cost.

## C   Evaluation

We instruct the LLM to give its answer following the format *'Final Answer': answer*. For AIME 2024 and AMC we parse the final answer from the output of the LLM and compare it against the ground truth answer (numerical values) and calculate the accuracy of the predictions. For MATH500, which requires more elaborated answers, we evaluate responses using an LLM-as-a-judge approach [Zheng et al., 2023] using GPT-4.1 as a judge of the answers from the LLM. We give to the judge the parsed answers from the LLM together with the ground truth and instructs it to say whether the parsed answer is correct or incorrect. The prompt used for the judge is as follows:

---
**Evaluation Prompt (LLM-as-a-judge)**

The following two expressions are answers to a math problem. They can be given as direct numerical answers or as a full reasoning. You have to judge whether they are equivalent. Only perform trivial simplifications, but accept numerical answers which are correct within a reasonable numerical tolerance.

Examples:

Expression 1: $2x + 3$
Expression 2: $3 + 2x$

Yes

Expression 1: 3/2
Expression 2: 1.5

---

Yes

Expression 1: $x^2 + 2x + 1$
Expression 2: $y^2 + 2y + 1$

No

Expression 1: $x^2 + 2x + 1$
Expression 2: $(x + 1)^2$

Yes

Expression 1: 3245/5
Expression 2: 649

Yes
(trivial simplifications are allowed)

Expression 1: 2/(-3)
Expression 2: -2/3

Yes
(trivial simplifications are allowed)

Expression 1: 72 degrees
Expression 2: 72

Yes
(give benefit of the doubt to units)

Expression 1: 64
Expression 2: 64 square feet

Yes
(give benefit of the doubt to units)

—

YOUR TASK

Respond with only "Yes" or "No" (without quotes). Do not include a rationale.

Expression 1: %(expression1)s
Expression 2: %(expression2)s

### C.1 Example Execution Trace

In this section, we consider an example question, taken from MATH500, answered both using the baseline LLM as well as with our cognitive tool pipeline. In particular, we provide the conversation trace obtained with the relatively small model Llama3.1-8B Instruct, to show how these tools help guiding the reasoning of smaller models.

The question that is passed as user input is as follows:

> Solve the math problem: 'Find the greatest common divisor of 3339, 2961, and 1491.'

The actual correct answer is **21**.

**Native response**

To find the greatest common divisor (GCD) of 3339, 2961, and 1491, we can follow these steps:

1. Find the GCD of 3339 and 2961:

We can use the Euclidean algorithm to find the GCD.

3339 = 2961 * 1 + 378
2961 = 378 * 7 + 375
378 = 375 * 1 + 3
375 = 3 * 125 + 0

The GCD of 3339 and 2961 is 3.

2. Now, find the GCD of 3 and 1491:
We can also use the Euclidean algorithm to find the GCD.

1491 = 3 * 497 + 0

The GCD of 3 and 1491 is 3.

Since the GCD of 3339 and 2961 is 3, and the GCD of 3 and 1491 is 3, the GCD of 3339, 2961, and 1491 is also 3.

So, the greatest common divisor of 3339, 2961, and 1491 is 3.

**Conversation trace with cognitive Tools**

To begin, I will call the 'understand_question' tool to break down the math question and provide more clear and detailed explanations of what's being asked and what's needed.

```
print(understand_question({
   'question': 'Find the GCD of 3339, 2961, 1491',
   'model': 'math_problem'
}))
```

## Understand Question Tool

Observation:
**Problem Analysis:**

1. **Core Mathematical Concepts:** The problem involves finding the greatest common divisor (GCD) of three numbers, which is a fundamental concept in number theory.

2. **Extracted Symbols, Variables, and Functions:**
* Variables: 3339, 2961, and 1491 (the three numbers for which we need to find the GCD)
* Function: GCD (greatest common divisor)

3. **Rephrased Problem into a Step-by-Step Sequence:**
• Find the prime factorization of each number: 3339, 2961, and 1491.
• Identify the common prime factors among the three numbers.
• Multiply the common prime factors together to find the greatest common divisor (GCD).

4. **Known Theorems or Techniques:**
* The Euclidean algorithm can be used to find the GCD of two numbers. We can extend this algorithm to find the GCD of three numbers by finding the GCD of the first two numbers and then finding the GCD of the result and the third number.

* The prime factorization of a number can be found using the fundamental theorem of arithmetic.

**Step-by-Step Instructions for the Upstream System:**

1. Find the prime factorization of 3339.
2. Find the prime factorization of 2961.
3. Find the prime factorization of 1491.
4. Identify the common prime factors among the three numbers.
5. Multiply the common prime factors together to find the greatest common divisor (GCD).
6. If necessary, use the Euclidean algorithm to find the GCD of the first two numbers and then the GCD of the result and the third number.

Please proceed with the first step: finding the prime factorization of 3339.

---

To continue solving the problem, I will call the 'use_code'.

```python
use_code(params...)
```

---

## Use Code Tool

Observation: Generated code block:

```
import math

def gcd_three_numbers(a, b, c):
    # Find the GCD of two numbers using the math.gcd()
        function
    gcd_ab = math.gcd(a, b)

    # Find the GCD of the result with the third number
```

```
      gcd_abc = math.gcd(gcd_ab, c)

      return gcd_abc

# Define the numbers
a = 3339
b = 2961
c = 1491

# Compute and print the GCD
result = gcd_three_numbers(a, b, c)
print("The GCD of", a, b, "and", c, "is:", result)
```

Execution Output: The GCD of 3339 2961 and 1491 is: 21

---

To continue solving the problem, I will call the 'examine_answer' tool to verify the correctness of the generated code and answer.

```python
examine_answer({'question': 'Find the greatest common divisor
    of $3339$, $2961$, and $1491$', 'current_proposed_answer'
    : 'The GCD of 3339 2961 and 1491 is: 21', 'model': '
    math_problem'})
```

---

## Examine Answer Tool

Observation:
### **1. Understanding the Problem**

The problem is to find the greatest common divisor (GCD) of three numbers: 3339, 2961, and 1491. The proposed solution uses the Euclidean algorithm to find the GCD of the first two numbers and then the GCD of the result with the third number.

### **2. Verifying the Given Solution**

The proposed solution is a Python function 'gcd_three_numbers' that takes three numbers as input and returns their GCD. The function uses the 'math.gcd()' function to find the GCD of two numbers and then applies the Euclidean algorithm to find the GCD of the result with the third number.

**Step-by-Step Breakdown:**

1. The function 'gcd_three_numbers' takes three numbers 'a', 'b', and 'c' as input.
2. It finds the GCD of 'a' and 'b' using the 'math.gcd()' function and stores it in 'gcd_ab'.
3. It finds the GCD of 'gcd_ab' and 'c' using the 'math.gcd()' function and stores it in 'gcd_abc'.
4. It returns 'gcd_abc' as the GCD of the three numbers.

**Logical Consistency and Mathematical Correctness:**

The proposed solution is mathematically correct and logically consistent. It uses the Euclidean algorithm to find the GCD of the three numbers, which is a well-established method.

**Edge Cases:**

The proposed solution does not explicitly handle edge cases such as zero or negative input values. However, the 'math.gcd()' function is designed to handle these cases correctly.

### **2.a) Testing and Validation**

**Test Cases:**

To verify the correctness of the proposed solution, we can test it with various input values.

```python
# Test case 1: GCD of 3339, 2961, and 1491
a = 3339
b = 2961
c = 1491
result = gcd_three_numbers(a, b, c)
print("Test case 1:", result == 21)  # Expected output: True

# Test case 2: GCD of 0, 0, and 0
a = 0
b = 0
c = 0
result = gcd_three_numbers(a, b, c)
print("Test case 2:", result == 0)  # Expected output: True

# Test case 3: GCD of 0, 0, and 10
a = 0
b = 0
c = 10
result = gcd_three_numbers(a, b, c)
print("Test case 3:", result == 10)  # Expected output: True

# Test case 4: GCD of -5, 5, and 10
a = -5
b = 5
c = 10
result = gcd_three_numbers(a, b, c)
print("Test case 4:", result == 5)  # Expected output: True
```

**Validation:**
All test cases pass, indicating that the proposed solution is correct.

### **3. Suggesting Improvements**

The proposed solution is already mathematically correct and logically consistent. However, we can suggest a few improvements:

* Add explicit error handling for edge cases such as zero or negative input values.
* Consider using a more robust method for finding the GCD, such as the Extended Euclidean algorithm.
* Add comments to the code to improve readability and understanding.

### **4. Providing a Judgment**

Based on the verification and testing, the proposed solution is **correct**.

Based on the previous steps, the proposed solution is correct. The greatest common divisor (GCD) of 3339, 2961, and 1491 is indeed 21. Therefore, I can provide the final answer.

**ANSWER: 21**

