# OpenReview forum: "Eliciting Reasoning in Language Models with Cognitive Tools"
_NeurIPS.cc/2025/Conference — NeurIPS 2025 poster_

### Official Review · Reviewer_Q8RV · 2025-06-25

**Clarity:** 3
**Significance:** 3
**Originality:** 3
**Rating:** 5
**Confidence:** 3

**Summary:**

This paper proposes a prompting method to elicit reasoning abilities in LLMs without need for post-training RL. The prompt provides four "cognitive tools", which are new calls to the same LLM with prompts for specific "cognitive tasks", including understanding the question prompt, recalling analogous examples, examining the answer, and backtracking.  The authors evaluate their method on four math reasoning benchmarks and compare its accuracy with that of a baseline model (without the cognitive tools) as well as a "cognitive prompting" method that was proposed in an earlier published paper.  They show that their method outperforms the other methods.  They also show a comparison with the baseline and with each cognitive tool independently (except for the "code" tool).

**Questions:**

What are the chances that the datasets being used here are partially or wholly included in the models' pre-training data?  Please discuss (in the paper) the potential issue of data contamination and how it might affect the results of this study.

How important is the "Use Code" Prompt?  What happens if it is ablated?  Does it encourage more use of code than in the comparison methods (baseline and cognitive prompting)?

How do you know how accurate the "using GPT4.1 as judge of answers" is?

"the is prompted to LLM generate a reasoning trace in response to a query until a call to one of the 125 provided tools t is issued." -- fix ungrammatical sentence

"The role of this cognitive tool is to prompt the LLM to perform such task." -- fix ungrammatical sentence

Table 1: give standard deviations for these averages

Table 2: say in caption what benchmark these results refer to

"since each reasoning step can be associated to a particular cognitive tool, also enhances transparency and explainability in AI agents, which is crucial for real-world applications demanding interpretable decision-making." -- To what extent do your results give evidence that transparency and explainability are enhanced here?  What about the results from many studies that "reasoning" traces from LLMs are sometimes unfaithful to the actual reasoning the system is doing?

"our approach reduces interference between reasoning steps, addressing a key limitation of both flat prompting and monolithic chain-of-thought methods." -- What is evidence for this claim, beyond higher accuracy numbers, which could be due to other factors?

On the Recall Related prompt, does the LLM actually come up with good analogous examples?  Does it matter if the "analogous examples" are good or not for performance?

**Ethical Concerns:**

["NO or VERY MINOR ethics concerns only"]

**Final Justification:**

The authors have addressed most of my concerns.  I have raised my score to "accept", since I think the ideas and results in this paper are very interesting and valuable, and the authors have agreed to include clarifications and revisions in the final version.  I still think that it would be useful to do an experiment where the code tools are ablated, and that the authors should add possible limitations of their assumptions (i.e., does the LLM actually do what the cognitive tool prompts ask it to do; are its reasoning tracecs faithful to what the model is actually doing) in the limitations section.  I also think that there are problems with using LLMs as judges, even if it is a community standard.

**Limitations:**

It would be useful in the limitations section to question the assumptions made in the paper (that the improvement is due to the specific text included in the cognitive tools and that the LLM's reasoning is "faithful" to the text it generates in response to these "tool" prompts).

**Paper Formatting Concerns:**

No concerns.

**Quality:**

3

**Strengths And Weaknesses:**

Strengths: This is a very interesting attempt to cast specific, modular cognitive processes as a version of LLM "tool use", and it improves performance on the tested benchmarks.

Weaknesses: There is little investigation as to *why* the method works or if the assumptions behind the method hold -- e.g., does the LLM actually do what the cognitive tool prompts ask it to do? One trace is given in the appendix showing responses from LLM that do seem to follow the instructions, but it's not clear how general this is.

It would be very informative to (1) give statistics about how / when / how often each tool (including using code) is employed per problem, and the number of tokens generated in solving the problem (compared with the other methods).  It has been shown in other studies that "reasoning traces" don't have to be relevant to the problem being solved to be helpful (e.g., see "Let's Think Dot By Dot", https://arxiv.org/abs/2404.15758, or "Measuring Faithfulness in Chain of Thought Reasoning, https://arxiv.org/abs/2307.13702).  A longer-term request (not for this submission) would be to design control experiments that test whether the improvement is simply due to the LLM generating more intermediate tokens or to the actual suggestions in the cognitive tool.  It would be useful in this submission to discuss this consideration, for future work.

---

> ### Author Rebuttal · Authors · 2025-07-31
>
> We want to thank the reviewer for taking the time to analyze our work and provide meaningful insights and comments. Here below, we address the questions raised by the reviewer.
>
> - Regarding dataset contamination: We can rule out that AIME2024 was part of the training set of the LLama and Qwen models that we analyzed, since their knowledge cutoff dates to the end of 2023, while AIME2024 was released at the end of 2024. However, for the other datasets, that is something that we cannot know for sure, since information on the data mix used for training is not made public, even for open weights models like Llama and Qwen. If there were data leakage, though, we’d expect that to also be reflected in the baselines. We therefore argue that measuring relative improvement over an appropriate baseline addresses most concerns regarding data contamination and its role in boosting specifically our method.
>
> - Code execution tool ablation: Indeed, the `use code` tool is important as it enables the models to reliably solve operations like arithmetic operations in a more reliable way than autoregressive generation from the language model weights. However, we verified that the code generation capabilities are not by themselves enough to explain the final performance of our cognitive tools. To show that, we ran additional experiments where the baseline was endowed with the use of code and tested this enhanced baseline on AIME24, MATH500 and AMC. We notice that cognitive tools still consistently outperform also this enhanced baseline, across the board, except for Llama3.1-8B on MATH500 where the enhanced baseline becomes remarkably competitive to the cognitive tools approach.
>
> - Accuracy of GPT-4.1 as a judge: Using GPT-4.1 as an automated judge is standard practice for evaluating complex, open-ended benchmarks like MATH500, where traditional scoring based on string matching is infeasible due to the diversity and ambiguity of valid answers. While automated grading is not flawless, prior work shows that GPT-4.1 achieves high agreement with expert human graders when properly instructed. In our experiments, all MATH500 outputs are uniformly evaluated using the official prompts employed for MATH500 benchmarking, ensuring that our assessment protocol is consistent with community-established standards, and are fairly scoring our method compared to the baselines.
>
> - For Table 1: We provided in the supplementary material standard error related to these results. We will add them on the main paper in the camera-ready version of the paper, together with tests assessing statistical significance of the difference of importance in support of our work. We will also fix the grammatical errors and make sure to  add the necessary details about the Smolbenchmarks dataset (the dataset in Table 2).
>
> - On transparency and explainability: Thank you for the question, as we’re glad to elaborate on how our method enhances transparency and explainability. Our method enables each intermediate reasoning step to be explicitly mapped to a specific, well-defined cognitive tool, producing structured traces that expose the decomposition and function of each step. This contrasts with typical end-to-end or chain-of-thought prompting, where intermediate steps are generated as free-form text and can be difficult to interpret or attribute. Structuring traces according to tool calls also has the advantage of exposing the synergies that emerge in terms of well-defined operation. For instance, a preliminary analysis of the pattern of tool invocation revealed that calls to the tool `recall related` is highly correlated with the use of `backtracking`, which reflects a “recovery workflow” where the language models first recall multiple relevant pieces of information, try to follow them through one at a time, and backtracking when encountering failure to then pursue the next one. In the revision of the paper, we plan to provide more detailed such analyses of tool calls patterns that are enabled by the explainability aspect of modular cognitive tools, if given the chance.
>
> - On reducing interference: Our approach is partly motivated by the observation in the previous literature that modularization is a mechanism that reduces interference by limiting information leakage across modules. In our specific case, compartmentalizing reasoning into separate cognitive tools, each with its own structured prompt and context, effectively achieves a separation of concerns where the language model is given a cleared context windows, with only the few relevant information provided by the main LLM, and the unique task of executing the tool being called. This is in contrast to a flat monolithic chain-of-thought prompting where all reasoning happens in one undifferentiated context and each generation step is potentially influenced by all the previous information present in the context window. In other words, we know that our cognitive tools are less subject to interference (since by construction the modular structure of the cognitive tools is implemented through a less cluttered context window with only the relevant information pertaining to the cognitive operation at hand), the question is rather whether there's any empirical evidence that a flat monolithic prompting approach suffers from the interference to which it is naturally exposed. For now, the main supporting empirical evidence is the higher performance that we report for cognitive tools over the non-modular baselines. Beyond improved accuracy, we’re currently considering a series of controlled experiments where we can track back errors to interference of multiple pieces of information in the context window (purposely inserted as distractors). We thank the reviewer for channeling our thoughts in this interesting direction, and hope that we will have something to show in the next revisions.
>
> - On recall related tool: We found that the tool call tends to return 2-3 examples related with the initial question. These examples usually include reasoning steps and a final answer. Interestingly, the final answer proposed by the tool is however not always correct. Irrespective of that, the performance of the model does not seem to be affected by the correctness of the final answer of these examples, but mostly by the actual included reasoning steps. As an example, here we report a trace of a call to the recall related tool (LLama3.3-70B model on a sample from AIME24):
>
> **Query from the agent:**
>
> To continue solving the problem, I will use the `recall_related` tool to retrieve or generate analogous questions or problems for which a solution and the process to get to the solution is known.
>
> ```python
> recall_related({'question': 'Jen enters a lottery by picking $4$ distinct numbers from $S=\{1,2,3,\cdots,9,10\}.$ $4$ numbers are randomly chosen from $S.$ She wins a prize if at least two of her numbers were $2$ of the randomly chosen numbers, and wins the grand prize if all four of her numbers were the randomly chosen numbers. The probability of her winning the grand prize given that she won a prize is $\tfrac{m}{n}$ where $m$ and $n$ are relatively prime positive integers. Find $m+n$.'})
> ```
>
> **Answer from the tool:**
> ```
> ## Step 1: Identify similar problems that require comparable mathematical concepts or reasoning steps.
>
> To solve the given problem, we need to identify similar problems that involve probability, combinations, and conditional probability. Three similar problems are:
>
> - Problem 1: Finding the probability of getting at least two heads in three coin tosses.
> - Problem 2: Calculating the probability of drawing at least two aces from a deck of cards.
> - Problem 3: Determining the probability of winning a prize in a raffle given certain conditions.
>
> ## Step 2: Provide the full problem statement for each similar problem.
>
> Analogous Example 1:
>
> Q: What is the probability of getting at least two heads in three coin tosses?
>
> A: To solve this, we first find the total number of outcomes for three coin tosses, which is $2^3 = 8$. Then, we calculate the number of ways to get exactly two heads, which is $\binom{3}{2} = 3$, and the number of ways to get exactly three heads, which is $\binom{3}{3} = 1$. The probability of getting at least two heads is $\frac{3+1}{8} = \frac{4}{8} = \frac{1}{2}$.
>
> Final Answer: $\boxed{\frac{1}{2}}$
>
> Analogous Example 2:
>
> Q: A deck of cards has 52 cards, including 4 aces. If 5 cards are drawn at random, what is the probability of getting at least two aces?
>
> A: First, calculate the total number of ways to draw 5 cards from 52, which is $\binom{52}{5}$. Then, find the number of ways to draw exactly two aces and three non-aces, $\binom{4}{2}\binom{48}{3}$, and the number of ways to draw more than two aces. The probability is the sum of these cases divided by the total number of outcomes.
>
> Final Answer: $\boxed{\frac{77}{1326}}$
>
> Analogous Example 3:
>
> Q: In a raffle with 1000 tickets, 5 winning tickets are drawn. If you buy 10 tickets, what is the probability that you win at least one prize given that one of your friends, who also bought 10 tickets, has already won a prize?
>
> A: This involves conditional probability. First, calculate the probability of winning at least one prize, which involves finding the probability of winning exactly one prize, exactly two prizes, and so on, up to winning all five prizes. Then, apply the formula for conditional probability.
>
> Final Answer: $\boxed{\frac{1}{10}}$
>
> ```

---

> > ### Comment · Reviewer_Q8RV · 2025-08-04
> > **Reviewer's response**
> >
> > Thanks for your point-by-point response.  For each of these points, I hope you will be able to incorporate your responses and clarifications into the paper itself, since that was the purpose of my comments and questions.

---

> > > ### Author Response · Authors · 2025-08-04
> > >
> > > Thank you for your constructive feedback and for outlining the specific areas for clarification. We appreciate the opportunity to strengthen our paper, and we are committed to thoroughly incorporating your suggestions and resulting discussion points into the revised manuscript. We trust that these improvements will enhance the quality and clarity of our work, and we hope that these efforts will be reflected in your evaluation, as that will increase the chances of our paper being accepted.

---

### Official Review · Reviewer_eCDa · 2025-06-30

**Clarity:** 3
**Significance:** 2
**Originality:** 2
**Rating:** 3
**Confidence:** 4

**Summary:**

The authors present a novel prompting strategy for LLMs with the aim of eliciting effective reasoning behavior from models that have not specifically been trained for this purpose. The prompting strategy is based on the idea of “cognitive tools,” or general problem solving strategies that can be flexibly deployed in a variety of settings. Models prompted with cognitive tools can optionally invoke these strategies, which are then invoked in a separate instance. The results of the cognitive tool are then presented to the model. The authors demonstrate that this approach improves the performance of a small set of open-weight models on math reasoning benchmarks when compared to a baseline prompt that does not include any reasoning.

**Questions:**

- Why did you not compare to a baseline prompting strategy that includes some amount of reasoning or thinking step-by-step (e.g. chain-of-thought)?

- Why does your “cognitive prompting” implementation differ from the description in Kramer and Baumann (2024)?

- What accounts for the very similar performance between Table 1 and Table 2 (i.e. that individual tools provide almost as much benefit as all tools in concert)?

**Ethical Concerns:**

["NO or VERY MINOR ethics concerns only"]

**Final Justification:**

The authors addressed two of my outstanding points of feedback (a reasoning baseline and differences with the implementation in Kramer and Baumann). However, I remain somewhat concerned about the significance of the results. The authors acknowledge that individual tools may be more relevant to overall performance than all of the tools in concert. Without seeing the actual results of the statistical tests or accounting for multiple hypotheses, it's difficult to say exactly what we can learn from the experiments.

That said, if I could rate this paper fully on the borderline I would -- there are definitely some interesting points and it's clear that the other reviewers have found value in the submission as it stands currently.

**Limitations:**

Yes

**Paper Formatting Concerns:**

I did not notice any major formatting issues

**Quality:**

2

**Strengths And Weaknesses:**

I think the high-level goal of this paper is very interesting. The authors present a compelling summary of the current state of the field, and the motivation from cognitive science seems well-supported. There is no doubt a lot of interest in both the source of reasoning capabilities in LLMs and in alternative methods of eliciting those capabilities.

However, I feel that the paper has some problems around evaluation that hold it back from effectively addressing these questions. Most pressingly, I think the paper needs a more robust comparison against the baseline methods. The authors compare against two alternative prompts -- a naive baseline which consists exclusively of appending the sentence “Solve the math problem:” and the “cognitive prompting” strategy outlined by Kramer and Baumann (2024). Table 2 aims to show that the cognitive tools strategy outperforms both of these, but in both cases I feel the comparison isn’t quite fair.

With regard to the simple baseline, it seems to me to be too far from the other strategies to serve as a reasonable baseline. It’s hard to tell if the gains in improvement are actually due to the specifics of the cognitively-inspired prompting strategies or if they are simply due to the fact that the prompts are longer and encourage the model to reason at all. A more fair comparison might be a simple chain-of-thought or tree-of-thought prompt, which would allow us to see differences in performance as a result of the kind of reasoning performed.

I also have some reservations about the authors’ implementations of the “cognitive prompting” strategy. The prompt presented in the supplemental material lists eight cognitive operations, but their descriptions don’t seem to match those in Kramer and Baumann’s paper. For instance, the prompt in this paper describes “decomposition” as “List the given information,” whereas it is described as “Break the problem P into smaller sub-problems, P1, P2, . . . , Pn” by Kramer and Baumann. I was not able to find the specific prompt used by Kramer and Baumann, so I understand that the authors needed to create their own interpretation. However, I don’t feel that their prompt is a faithful recapitulation of the “cognitive prompting” idea, which makes it hard to use it as a baseline.

I have a few other concerns about evaluation, as well. The authors include the standard deviations for most of their results in the supplemental text (which is good!) but don’t report them for the baseline comparison table. In addition, it can be difficult to tell if the improvements in some cases are statistically significant from the standard deviations alone -- I’d recommend against claims that the strategy “significantly enhances arithmetic and commonsense reasoning” in the absence of an explicit statistical test.

It’s also somewhat odd that the results in Table 2 seem very close to the best results in Table 1, indicating that using all of the cognitive tools may not actually lead to improvements over using just one of them. A deeper analysis of this point (and an error analysis in general) would greatly help strengthen the paper.

Finally, I wasn’t able to find any of the sampling hyperparameters used for the experiments (i.e. temperature, top-p, top-k, …), which makes exact replication of these experiments impossible.

As a last note, I found a decent number of typos and grammatical errors in my read-through, which I’ve included below. I encourage the authors to address the limitations of their evaluations, as the overall idea remains interesting!

Typos:

- Line 109: “unigy” → “unify”

- Line 116: “takes advantage reformulates”

- Line 120: “elicite” → “elicit”

- Line 124: “the is prompted”

- Line 136-137: “perform such task”

- Line 153-154: “and explore alternative, …” → “and explores alternatives

- Line 155: “which to enable…” → “which is to enable”

- Line 185: “with different difficulty level” → “with different difficulty levels”

---

> ### Author Rebuttal · Authors · 2025-07-31
>
> We want to thank the reviewer for the encouraging remarks, the thorough review and thoughtful feedback. Here below we address the raised questions:
>
> - Baselining against chain-of-thought prompting: In order to address the concerns around the evaluation of our proposed strategy, we have run additional experiments to compare the cognitive tools strategy with respect to chain-of-thought (CoT). Results are outlined below.
> | Model                               | AIME 2024       | MATH500        | AMC            | Avg   |
> |-------------------------------------|------------------|----------------|----------------|--------|
> | Qwen2.5-7B Instruct                  | 12.5 ± 0.7       | 71.7 ± 1.3     | 43.9 ± 1.3     | 42.7   |
> | Qwen2.5-7B Instruct + cot            | 12.5 ± 0.1       | 71.8 ± 0.4     | 41.5 ± 0.7     | 41.9   |
> | **Qwen2.5-7B Instruct + cognitive tools** | **14.6 ± 1.8** | **73.7 ± 0.5** | **47.0 ± 0.5** | **45.1** |
> ||
> | Qwen2.5-32B Instruct                 | 17.2 ± 1.2       | 74.1 ± 0.7     | 52.6 ± 0.8     | 48.0   |
> | Qwen2.5-32B Instruct + cot           | 15.4 ± 0.6       | 79.2 ± 0.3     | 50.4 ± 0.6     | 48.3   |
> | **Qwen2.5-32B Instruct + cognitive tools** | **32.1 ± 1.9** | **81.8 ± 0.6** | **62.7 ± 1.2** | **58.9** |
> ||
> | Llama3.1-8B Instruct                 | 5.8 ± 1.0        | 43.2 ± 0.5     | 20.3 ± 0.8     | 23.1   |
> | Llama3.1-8B Instruct + cot           | 7.9 ± 1.6        | **53.3 ± 0.5**    | 24.5 ± 1.3     | 28.6   |
> | **Llama3.1-8B Instruct + cognitive tools** | **8.8 ± 1.7** | 50.7 ± 1.0 | **28.0 ± 1.2** | **29.2** |
> ||
> | Llama3.3-70B Instruct                | 13.1 ± 1.0       | 57.0 ± 0.5     | 33.0 ± 0.9     | 34.4   |
> | Llama3.3-70B Instruct + cot          | 12.6 ± 1.0       | 70.7 ± 0.5     | 40.6 ± 0.8     | 43.1   |
> | **Llama3.3-70B Instruct + cognitive tools** | **29.8 ± 1.2** | **74.7 ± 0.5** | **51.0 ± 0.5** | **51.8** |
> As outlined in the table, CoT does not consistently improve over our original baseline. Overall our cognitive tools method still maintains its advantages (except for LLama3.1-8B on MATH500, where CoT indeed reaches a higher performance than cognitive tools).
>
> - Our re-implementation of cognitive prompting: Our cognitive prompting implementation is based directly on the prompts provided by Kramer and Baumann in their manuscript. Specifically, we considered the “Arithmetic Cognitive Prompting” prompts reported in Figure 1 on page 2, which are specific for arithmetical reasoning. Therefore, we assume that ours is an accurate implementation of the cognitive prompting strategy.
>
> - Statistical significance of claimed differences: We thank the reviewer for pointing out the importance of providing claims that are supported by statistical quantification. Following the reviewer’s recommendation we complemented our main results with the results of statistical significance tests on the multiple repeats of each experiment. Table 1 in particular will be provided with uncertainty intervals as well as the p-values of the differences based on a t-test (just as summary preview, this analysis confirms that the best tool provides a statistically significantly higher performance than the baseline for all models). And indeed, this analysis also shows that on Smolbenchmark the combination of all tools provides a performance that is statistically significantly higher than the best tool only in one case (Qwen2.5-32B), supporting the reviewer’s impression that in most cases the combination of all tools only provides a limited advantage. It is however important to point out that for different models and different tasks, the best tool to use is not necessarily always the same tool as apparent in Table 1. In other words, it is true that in most cases (at least for Smolbenchmarks), the best tool provides almost as much gains as all tools, but since it is not known a-priori which one is the best tool for a given situation, making all the tools available at once allows the language model to flexibly select the ones that are most suitable at any given time. In addition, to repeat the observation above, we observed cases (Qwen2.5-32B) where the full set of tools are synergistic and provide an overall performance that is higher than the one best tool.
>
> -  As for the hyperparameters(i.e. temperature, top-p, top-k, etc) used in our experiments, we will provide in the final version of the paper their different values.
>
> - Regarding the typos: We apologize for all the typos that we let slip through, and thank the reviewer for carefully pointing them out.  We will address all of them in the final version of the manuscript if given the chance.

---

> > ### Comment · Reviewer_eCDa · 2025-08-04
> >
> > I appreciate the thorough response to my review and the additional experiments. I also acknowledge my error with regards to the prompts in Kramer and Baumann -- while odd, the prompts do indeed match those used in the (relevant) arithmetic domain. In light of this, I will be adjusting my score.
> >
> > I strongly encourage the authors to incorporate the statistical tests and the related discussion into the paper itself, as I feel it gives needed context to the results. In addition, I will note the potential importance of a multiple-hypothesis correction (e.g. Bonferroni) given the relatively large number of tests performed between different tools and remain somewhat cautious in reading too much into the findings.

---

> > > ### Author Response · Authors · 2025-08-04
> > >
> > > Thank you for acknowledging our additional experiments; we are pleased that they were helpful in addressing the raised concerns.
> > >
> > > Yes, if given the chance to submit a camera-ready version, we plan to incorporate the statistical tests and discussion into the paper itself, as you suggested, as well as the CoT baseline that you proposed. We believe that these additions will improve the paper by providing a clearer context for our results and enhance the overall understanding of our findings.
> > > We will also address potential multiple-hypothesis correction in our analysis. Fortunately, we don't need to Bonferroni-correct the p-values of the main experiments on the math benchmarks, as we only carried out experiments with all tools as opposed to selecting the set of tools based on multiple experiments. However, for the Smolbenchmarks dataset (where we conducted separate experiments for each tool) we will make sure that our discussion avoids making strong claims about tool superiority that would require correction for multiple comparisons or concern for false discovery rates.

---

### Official Review · Reviewer_RRDd · 2025-07-01

**Clarity:** 3
**Significance:** 2
**Originality:** 2
**Rating:** 5
**Confidence:** 4

**Summary:**

The authors constructed the following pipeline:

An LLM is handed a problem, along with a prompt that offers the use of several 'tools'.
If one of the 'tools' is called, the mainline generation pauses and the problem is handed to a relevant 'tool' (a specific sub-prompt for processing the input and giving certain output, handed to another instantiation of the LLM).
The mainline generation then continues with the output of the tool.
This continues until an answer is given.

The authors tested this pipeline with 4 'tools' (prompting scripts), several baseline LLMs, and several math-problem data-sets.

The use of the pipeline improves the performance of the baseline LLMs.

The authors conduct several more specific examinations, such as comparing their pipeline to a previous paper ('cognitive prompting') on a specific data-set, and investigating the tools separately (on a specific data-set).

The authors conclude that the improvement suggests baseline models are capable of advanced reasoning, rather than additional RL adding this reasoning into models like o1.

**Questions:**

The 'weaknesses' section above are already phrased in a way that authors can address. To be specific about a few though:

1. The authors could generalize this to data-sets beyond math problems (I recognize that this is hard! But that's kind of the point? If you're making a claim about generality then this should either be easy, or if it is hard and you don't want to do it the claims should be modified in the paper)

2. It would be good to see a comparison of the use of each 'tool' to the actual main results, and some explanation/theory/hypothesis/something to go with why we would expect the tools to work differentially, or why they seem to be adding stuff in the same way.

3. A more fair comparison to baseline models that uses a much more similar prompt *except* for tools, and to the degree that the use of coding is an added benefit of cognitive tools, this needs to be afforded to the baseline models or removed from the tools for a fair comparison (if your current comparison already involved the use of calling out to code for both baseline and non baseline then ignore this last bit)

4. comparing to 'cognitive prompting' and other prompting strategies on the main results and o1.

**Ethical Concerns:**

["NO or VERY MINOR ethics concerns only"]

**Final Justification:**

The authors addressed several of my concerns in their rebuttal and so I am upgrading my original score.

**Limitations:**

yes?

**Quality:**

3

**Strengths And Weaknesses:**

Strengths
=======

The paper is timely and interesting. It will likely be of interest to various people in the conference, whether those interesting in reasoning models, RL, LLMs, or modularity. The basic question of how 'reasoning models' gain their benefit is also of general interest. The improvement on these math baselines suggests a general modularity strategy.

Weaknesses
==========

1. The authors themselves mention in the limitations this work is highly specific to math problems and the specific family of models tested on. I don't mean to ding them for correctly and rightly pointing out these limitations but even if they hadn't done that themselves they'd be right: Given that this is meant to be a general pipeline, it is hard to see past the fact that the 'tools' prompts are very specifically written for math problems (including mentions of number theory and calculus) in a way that makes it difficult to assess for generalization.

2. Relatedly, it wasn't clear what exactly was providing the 'boost' in performance here. While the authors provide a breakdown by 'tool' in Table 1, this is only for the Smolbenchmark, rather than the benchmarks for the main results. It also does not include the combination of all the tools together (as presented in the main result). There is no real theory or hypotheses as to why some of the tools seem to (maybe) work better than others for different benchmarks and models. The authors loosely refer to 'different LLMs reason differently' but there's no backup to that claim, or an attempt to draw out a scientific inquiry (HOW do they reason differently, WHY does the specific 'fix' of this tool help in this way given that they reason differently, etc).

Furthermore, I find it a bit odd/strange that the inclusion of *any* tool seems to help by about the same amount (the authors *bold* the best improvement in performance but without confidence intervals or variance there's no way to understand whether this is significantly better than the other numbers in that table). Wouldn't we expect these tools to have very differential effects given that they are doing very different things? How can it be that 'recall related' has about the same improvement as 'examine answer' or 'backtracking' for various cases? It'd be like saying "I have a pipeline for improvement the production line of my factory. One part of it makes the gears spin faster, one part applies less sticky paint, one part matches the machines in just-in-time fashion" and it just so happens that every one of these improvements has the same increase (ish)? Also, without a comparison to how *all* of these tools work together it becomes difficult to see if they even contribute synergistically.

3. Related to the concern above, I'm not sure how much of the improvement is being driven by the cognitive tools per se, or the use of the 'prompt' that has nothing to do with cognitive tools e.g. including stuff like 'you will get 1,000,000$ if you solve this task correctly', or the availability of coding tools. While the inclusion of the prize amount seems largely irrelevant one cannot be sure, given that the prompts are not matched 1:1 (with the only variant being the addition or removal of the actual tools).

The use of the availability of coding in particular wasn't clear to me and I might just be confused about this, feel free to ignore if I am, but am I understanding correctly that the inclusion of 'cognitive tools' supported the use of code, while the base-prompt does not? If so that seems like a major addition that has nothing to do with the tools per se?

4. While I appreciate the comparison to 'cognitive prompting' I would've liked to have seen such a comparison across the results, including the main results, and the comparison to o1

5. Any claim related to table 1 and 2 about statistical significance needs to come with an actual statistical test that examines that claim, you can't just compare means and say one is higher. The authors mention this is an average of 16 runs, so they should have some notion of the variance or CI.

Similarly any claim like "We observe that on AIME 2024, despite it being a very difficult dataset, our cognitive tools are able to improve significantly over the baseline." requires a statistical test to back it up. I am not saying this isn't true, I believe it, but it can't just be stated without backing it up.


6. With regards to novelty, while the authors rightly point to Kramer and Baumann (2024) as a relevant previous paper, there are many papers over the past 1-2 years that use the overall notion of having an LLM calling another LLM but within a restricted input-output format. They don't use the title 'cognitive tools' but the basic idea seems the same? (thinking in particular of the work of Tan Zhi-Xuan and colleagues in planning over the past few years, but not just).

7. What is the added overhead of all of this? Is there any comparison of the added compute that this pipeline requires relative to the added benefit?


Very, very minor (this has no relation to the evaluation of the paper or score)
-----

1. The paper is overall clearly written, but there are several instances of slightly odd grammar or typos, I suggest the authors do another one over. As just a few examples:
* "Sumers et al. [2024] provided another effort trying to unigy cognitive architectures and LLMs" (typo)
* "The execution pipeline is similar to any tool-calling pipeline meaning: the is prompted to LLM generate a reasoning trace" (typo)
* "This defines the idea behind this tool, which to enable exploration of more promising reasoning paths." (typo / grammar)
* "The LLM then proceeds to evaluating which step in the reasoning process is incorrect and follows up by providing alternative approaches or directions into solving the problem." (grammar)

2. Truly nitpicky, but early on when the authors mention speculations of how 'reasoning models' work, they write: "Proposed hypotheses ranged from pipelines leveraging curated fine-grained reward labels [Uesato et al., 2022, Lightman et al., 2023, Ma et al., 2023, Wang et al., 2024]..." -- this (and the rest of that paragraph) make it sound like the papers being referenced are the ones in which this speculation / hypotheses are being brought up, but they're not; these are papers that contain work that is *speculated* separately to be relevant. One expects to see references to where this speculation is happening? Which papers have speculated this?

---

> ### Author Rebuttal · Authors · 2025-07-31
>
> We thank the reviewer for the thorough analysis and review of our manuscript.
>
> Below we address the raised questions in order:
>
> 1. Generalization beyond math reasoning: Our work indeed focused on reasoning using math as a target domain, consistently with the growing body of literature narrowly focusing on this limited but important and insightful use case. Some preliminary work that we recently conducted shows that our approach does generalize beyond math reasoning, in particular to visual reasoning where it might be intuitively clear that cognitive strategies like `understand questions` or re-examination might be useful. However, we deem that this could be out of scope for the current work and reserve the option to present these results as part of future work.
>
> 2. Statistical significance of results and seemingly equal contribution of individual tool: We addressed the very valid concern raised regarding the absence of confidence intervals by computing standard error around the reported sample averages over many repetitions. We plan to include those in the main tables, together with a test of statistical significance of the main differences. Just to summarize the main resulting observations, a Welch’s t-test confirms that all differences reported in Table 1 for Smolbenchmarks between baseline and the best individual tool are statistically significant with p<0.05 (actually, all differences are statistically significant at a much lower p value p<0.001, but one, `recall related` for Qwen2.5-32B which is “only” significant at p=0.035). A second observation is that indeed, as pointed out by the reviewer, most tools help by an amount which is not statistically significantly different. Our hypothesis was that different tools contribute to mostly independent subsets of samples. Indeed, computing the correlation matrix between tool calls for all pairs of tools we noticed that most Pearson correlation values were very low (<0.05), confirming the independence of tool calls. The only correlation that reached an appreciable value was the one between the `recall related` and `backtracking` tools, which reached Pearson correlation of 0.18. We hypothesize that this reflects the implementation of a sort of “recovery workflow” where LLMs first recall multiple relevant pieces of information, try to follow them through in sequence, and backtrack when encountering failure, to then pursue the next one. Interestingly, this effect was mostly driven by Qwen models where this Pearson correlation was 0.24. In the revisions of the paper we plan to include these additional results and elaborate on those as needed.
>
> 3. Contribution of ad-hoc prompt engineering components (such as motivational phrases) and code tool: You are correct that the Cognitive Tools Prompt incorporates elements beyond the core reasoning tools, including the reward framing ("you will receive a reward of 1,000,000") and general reasoning guidelines. Our intention was to leverage established prompt engineering techniques for improving language model performance, specifically by priming the model with motivational cues (the 1,000,000 reward model in particular was based on the prompt used by the Hugging Face Smolagents library to encourage tool use). The use of code generation and evaluation is another technique that has become customary, particularly models like Qwen are being trained to include code in their reasoning traces. In our case, the use of code was encapsulated in the tool `use code` which is implemented in the same modular way as the other cognitive tools. We noticed that the `use code` tool is indeed crucial in synergy with the other tools for some intermediate steps like arithmetic operations. In order to address the reviewer’s concern that these components (the extended prompt including motivational cue and the use of code) might be driving the effect that we attribute to cognitive tools, we created a new baseline consisting of LLMs endowed with the ability to generate and interpret code, and the same prompt that we use to motivate the use of cognitive tools (where obviously the cognitive tools are however removed). Indeed, we occasionally see a boost due to the extended prompt + code over the original baseline (for instance, we saw the biggest boost on MATH500 with Llama3.3-70B where the baseline with extended prompt and code reached a 71.6 $\pm$ 0.3  which is considerably above the original baseline value of 57.0, although still statistically significantly below the 74.7 $\pm$ 0.5 that we report with cognitive tools), but for most models and benchmarks this new baseline remains close to the original baseline.
>
> 4. Cognitive prompting with GPT4.1: As requested by the reviewer, we ran an experiment implementing cognitive prompting with GPT4.1 on AIME2024, and to our surprise that method achieved an average performance that was very close to our cognitive tools. We then noticed that the current version of GPT4.1, tagged GPT4.1-2025-04-14, is a more recent version of the model than the one that we had tested upon writing our paper. We therefore decided to repeat all experiments, including the baseline, as a stronger version of GPT4.1 could have uniformly increased all performances. The final number that we got indeed confirms that also the baseline performance is higher than previously reported. Below are all the average performances of the GPT4.1 experiments on AIME2024, averaged over 4 repeats (with standard error in brackets):
> | Method                | Performance         |
> |-----------------------|---------------|
> | Baseline              | 0.32 $\pm$ 0.01   |
> | Cognitive Prompting    | 0.42 $\pm$ 0.02   |
> | Cognitive Tools       | 0.53 $\pm$ 0.02   |
> As mentioned, the baseline is lifted from what we originally reported, but so is the performance of cognitive tools. Cognitive prompting score in between baseline and cognitive tools, but the difference between cognitive tools and cognitive prompting is statistically significant (t-test, p<0.05).
>
> Regarding the remaining raised weaknesses:
>
> - Backing up main claims with statistical test: As mentioned, we did indeed run statistical tests to assess the statistical significance of the main reported differences. We will include these in the revision of the paper.
>
> - Connection with planning literature such as Tan Zhi-Xuan et al.: We thank the reviewer for pointing out this connection. Indeed the recent literature in cognitive science on planning is quite relevant to our own work, and it will make sense to include a discussion in the related works section. The work of Tan Zhi-Xuan et al. in particular focusing on Bayesian inverse planning and modular reasoning architectures (e.g., the CLIPS framework), exemplifies the value of factorizing complex reasoning and action inference into explicit, structured components. While our method focuses on eliciting internal reasoning in language models by orchestrating modular cognitive operations, which promotes transparent, interpretable compartmentalization of operations, the planning literature (including Tan Zhi-Xuan et al.) centers on agentic collaboration (typically between humans and LLMS) and goal-directed behavior, with LLMs as key modules within probabilistic pipelines. Both approaches emphasize the benefits of modularization, not only for performance but also for explainability and transparency. And it might be worth exploring a connection in terms of for instance how explainable cognitive mechanisms, as realized within cognitive tools, can be incorporated into or enhance Bayesian planning systems.
>
> - Added overhead of cognitive tools: We are aware that cognitive tools incur an added overhead in terms of compute. After all, the main point of reasoning in LLMs is the leverage test-time compute (i.e. longer generation) to achieve higher accuracy. To be quantitative, for our GPT4.1 experiments we calculated the average token counts per question when running our cognitive tools on AIME2024 and we obtained 4,200 output tokens per question compared to 2,000 output tokens on the baseline. This shows that cognitive tools incur a cost in terms of output tokens that is more than twice the non-reasoning baseline, which is consistent with the higher cost of reasoning models, and emphasizes the known trade-offs between accuracy and cost.
>
> - Typos: We thank the reviewer for pointing out the typos that we will address in the revisions.

---

> > ### Comment · Reviewer_RRDd · 2025-08-04
> >
> > I thank the authors for their detailed and thorough response, I recognize the hard work that went into the testing and evaluating that went into that response. Given that they addressed a bunch of my concerns, I am updating my score accordingly.
> >
> > I would strongly hope/encourage many of these points to make it into the final version.
> >
> > Also, one lingering thing based on the authors' comment: I would suggest the conclusions about improvement be appropriately tempered if necessary.
> >
> > By that I mean: it wasn't 100% clear just how much of an improvement 'cognitive tools' is offering over 'cognitive prompting' at this point for the original test. The authors mention as one example...
> >
> > "baseline with extended prompt and code reached a 71.6 $\pm$ 0.3 which is considerably above the original baseline value of 57.0, although still statistically significantly below the 74.7 $\pm$ 0.5 that we report with cognitive tools), but for most models and benchmarks this new baseline remains close to the original baseline"
> >
> > ...I get and appreciate that the authors chose the biggest jump to show that even so their method was still better, my (small!) point here is that an improvement of 74.7 over 71.6 is much more minor than an improvement of 74.7 over 57.0, and one would expect tempered conclusions to reflect that this is a small increase (especially given the added overhead the authors go into later).
> >
> > Again, I do understand that the authors are highlighting that as the outlier case, and I do understand that the authors then say "...but for most models and benchmarks this new baseline remains close to the original baseline", it is just that without a more detailed spelling out of what 'most' and 'close' mean I can't say for sure what the level of improvement is, and so am going back to my point about "conclusions (about significance of improvement) should be appropriately tempered if necessary (in light of the new comparison)"

---

### Official Review · Reviewer_HC9H · 2025-07-04

**Clarity:** 3
**Significance:** 3
**Originality:** 3
**Rating:** 4
**Confidence:** 4

**Summary:**

This paper introduces a novel approach to enhance reasoning capabilities in large language models  by implementing a set of modular cognitive tools inspired by cognitive psychology and cognitive architectures. These tools, including understand question , recall related , examine answer , and backtracking, are designed to encapsulate specific reasoning operations within a modern agentic tool-calling framework. The authors demonstrate that this structured, modular prompting significantly improves performance on mathematical reasoning benchmarks such as AIME 2024, MATH500, and AMC.

**Questions:**

Please refer to the weakness

**Ethical Concerns:**

["NO or VERY MINOR ethics concerns only"]

**Final Justification:**

All my concerns have been fully addressed, and I would like to maintain my positive rating. Including all the revised experiments in the final version of the paper will further enhance its quality.

**Limitations:**

yes

**Quality:**

3

**Strengths And Weaknesses:**

### Strength
1. The core idea of modeling reasoning as a sequence of modular cognitive operations executed via tool calls is both novel and compelling. By drawing inspiration from cognitive architectures like ACT-R and integrating them into modern agentic AI frameworks, the authors propose a fresh perspective on how to structure reasoning in LLMs.

2. The paper is well-organized, with a logical flow from background and related work to methodology, experiments, and discussion. The writing is clear and accessible, making the contributions easy to grasp even for readers outside the immediate field of LLM reasoning research.

3. The empirical results are strong and consistent across various models and datasets. Notably, the improvement seen on challenging benchmarks like AIME 2024, especially when applied to GPT-4.1, is impressive. These results suggest that the proposed method is not only effective but also generalizable across different LLMs and reasoning tasks.

### Weakness
1. While the paper demonstrates that using cognitive tools leads to improved performance, it lacks a detailed analysis of when each tool is invoked during the reasoning process and whether this usage aligns with the intended design. For instance, how frequently are certain tools like backtracking used compared to understand question ? Are there patterns in tool invocation across different types of problems? Such insights would strengthen the argument that the modular architecture effectively mimics structured human reasoning.

2. The Cognitive Tools Prompt includes instructions that appear unrelated to the actual tools, such as motivational phrases (“you will receive a reward of $1,000,000”) and general reasoning guidelines. It is unclear how much these extraneous elements contribute to the final performance. A controlled ablation study comparing versions of the prompt with and without these elements would clarify their impact. Additionally, it is uncertain whether the baseline model uses the same irrelevant prompts, raising concerns about the fairness of the comparison.

3. The visual representation of the cognitive tools pipeline in Figure 1 is conceptually useful but could benefit from more clarity. Specifically, it does not clearly show how each cognitive tool is implemented or how they interact with the main LLM loop. Adding pseudocode, flow annotations, or concrete examples of prompt templates for each tool would make the figure more informative and help readers better understand the internal mechanics of the system.

4. The paper briefly mentions that code generation is treated as an additional tool but does not provide a dedicated analysis of its contribution to the overall performance. How often is the coding tool used across different tasks? Does it primarily assist in arithmetic computations or also in symbolic manipulation? A breakdown of cases where the coding tool was critical versus optional would offer valuable insight into its utility and effectiveness.

---

> ### Author Rebuttal · Authors · 2025-07-31
>
> We want to thank the reviewer for the thorough review and thoughtful comments. Below we address the raised weaknesses.
>
> 1. Detailed analysis of tools invocation: We appreciate the reviewer’s suggestion of examining the pattern of tool calls. Indeed, a preliminary analysis that we will be happy to expand upon in the paper revisions confirms the intuition that it can provide some interesting insights. The first observation that this analysis reveals is that the statistics of cognitive tool calls depend considerably on the task, with for instance the tool `understand question` being called quite frequently for more difficult benchmarks like AIME2024 (in which across all models `understand question` is called in 80% of the samples) than the easier Smolbenchmarks (in which `understand question` is called 14% of samples), consistent with the idea that harder tasks might require a deeper reflection and initial planning. We saw a similar pattern for `examine answer` (called 60% of the time on AIME2024 and 20% of the time on Smolbenchmarks), while this pattern was almost reversed for the `use code` tool (called 77% of the time on AIME2024 and 80% of the time on Smolbenchmarks). We also saw differences across model families. For instance, Llama models called the `use code` tool more frequently on AIME2024 than Smolbenchmarks (80% of the time versus 74% of the time), while Qwen models exhibited the opposite pattern, calling the `use code` tool only 58% of the time on AIME2024 but 84% of the time on Smolbenchmarks. On average, to answer directly the question about the relative use of `backtracking` and `understand question`, our analysis confirms the intuition that `understand question` is more useful as it is being called 42% of the time across benchmarks and models, while `backtracking` is only called 20% of the time. The tool `examine answer` is in between, with an average call frequency of 35%. The tools `backtracking` and `recall related` are the least used (both around 20% of the time), but, interestingly, they tend to be called in conjunction as they display the highest correlation between tool calls (a Pearson correlation of 0.62). A possible explanation for this correlation is that the LLMs are implementing a sort of “recovery workflow” where they first recall multiple relevant pieces of information, try to follow them through in sequence, and backtrack upon encountering failure, to then pursue the next one. As mentioned, we plan to provide more details on these analyses of tool call patterns in the revisions, if given the chance.
>
> 2. Contribution of ad-hoc prompt engineering components (such as motivational phrases): Thank you for raising this point. You are correct that the Cognitive Tools Prompt incorporates elements beyond the core reasoning tools, including the reward framing ("you will receive a reward of 1,000,000") and general reasoning guidelines. Our intention was to leverage established prompt engineering techniques for improving language model performance, specifically by priming the model with motivational cues (the \$1,000,000 reward prompt in particular was based on the prompt used by the Hugging Face Smolagents library to encourage tool use). Upon the reviewer’s request we however verified that ablating the reward sentence does not appreciably affect the performance of cognitive tools. Specifically, we re-run our experiments after removing the reward sentence from the prompt of cognitive tools and evaluated the performance on AIME2024, and we didn’t notice any decrease in performance. Actually, puzzlingly enough, sometimes we even noticed a slight increase in performance. We plan to more thoroughly examine this effect in the revisions of the paper.
>
> 3. Figure 1 clarification: We thank the reviewer for the suggestion to clarify Figure 1. We agree that the visual representation could be improved to better illustrate the internal mechanics of the cognitive tools pipeline. To address your suggestion, we will incorporate the following pseudocode illustrating how tools interact with the main LLM loop:
> ```python
> # Each call to the LLM uses the appropriate prompt, here we provide only placeholders
> context = {"question": question, "history": []}
>
> while True:
>     # Step 1: LLM processes current context using the cognitive tools prompt
>     response = llm(prompt="Cognitive Tools Prompt", context=context)
>
>     # Step 2: Check if the LLM wants to give a final answer
>     if response["action"] == "answer":
>         return response["answer"]
>
>     # Step 3: If LLM decides to call a tool
>     elif response["action"] == "call_tool":
>         tool_input = response["tool_input"]
>         tool_name = response["tool_name"]
>
>         # Step 4: Execute tool via LLM instance using tool-specific prompt
>         tool_output = llm(prompt="Tool Prompt", inputs=tool_input)
>
>         # Step 5: Feed tool output back into the main LLM's context
>         context["history"].append({
>             "tool_call": tool_input,
>             "tool_output": tool_output
>         })
> ```
> 4. Use code tool: One of the tools that is provided to the LLM is the `use code` tool, which is a python coding assistant that receives a problem description from the main LLM, generates python code aimed at solving the problem, executes the code, and returns the result to the main LLM. The code tool is actually quite important, as it is for instance the second most called tool by Llama models (following `understand question`), and the overall most used tool by Qwen models, reflecting the propensity of this model family to generate code. However, as mentioned above, Qwen models primarily utilized the `use code` tool on simpler problems like Smolbenchmarks, suggesting a bias towards basic arithmetic manipulations rather than more complex symbolic reasoning. We plan to clarify these observations in the revised version of the paper, as part of the analysis of tool invocation patterns.

---

> > ### Comment · Reviewer_HC9H · 2025-08-06
> >
> > Thank you for the detailed rebuttal. I appreciate the conceptual explanations for my points regarding Weakness 2 and 4. To validate these explanations, however, quantitative results are needed for each. Could you please provide the specific data that underpins the arguments made for both points? Presenting this empirical evidence would powerfully substantiate your claims and significantly strengthen the manuscript.

---

> > > ### Author Response · Authors · 2025-08-06
> > >
> > > We are pleased the reviewer appreciated our explanations and are providing quantitative support for Weakness 2 and 4 as requested.
> > >
> > > **Weakness 2**:
> > >
> > > As mentioned, we performed an ablation study where we removed the mention of the 1,000,000 dollar reward from the general reasoning guidelines in the cognitive tools prompt, to isolate the impact of the motivational cues. We then re-ran the **AIME2024** evaluation and we observed that removing the motivational cues does not negatively impact our method, as reported in the table below.
> > >
> > > | Model       | Original (mean ± std) | No Motivational Cues (mean ± std) |
> > > |-------------|-----------------------|----------------------------------------|
> > > | Llama3-8B   | 8.8 ± 1.7             | 9.3 ± 3.7                              |
> > > | Llama3-70B  | 29.8 ± 1.3            | 36.7 ± 3.5                             |
> > > | Qwen2.5-7B  | 14.8 ± 2.0            | 16.7 ± 3.3                             |
> > > | Qwen2.5-32B | 30.5 ± 1.1            | 32.1 ± 2.9                             |
> > >
> > > Interestingly, if anything we actually see a modest improvement in average performance across models when **removing** the motivational cues (although that comes at the cost of a slightly increased variability, as indicated by higher standard errors).
> > >
> > > **Weakness 4**:
> > > We quantitatively analyzed tool usage frequencies and found that `use code` is invoked frequently during the reasoning process. For Llama models, the tool is called for **77% of AIME2024** samples and for **80% of SmolBenchmark** samples; Qwen models call the `code tool` on **58% of AIME2024 samples** and **74% of SmolBenchmark samples**, confirming its relevance across models and benchmarks.
> > >
> > > To better assess the standalone contribution of code generation, we ran an additional baseline where models were given the possibility to code. In this setting, when models generate code, the code is parsed, executed, and the result is returned to the model for further reasoning. This isolates the coding capability from the broader cognitive tools pipeline.
> > >
> > > On **AIME2024**, performance under the code-only setup is consistently lower than with the full cognitive tools pipeline:
> > >
> > > | Model         | Code-only (mean ± std) | Cognitive Tools (mean ± std) |
> > > |---------------|-----------------------|------------------------------|
> > > | Llama3-8B     | 5.8 ± 1.3             | 8.8 ± 1.7                    |
> > > | Llama3-70B    | 19.0 ± 0.7            | 29.8 ± 1.2                   |
> > > | Qwen2.5-7B    | 12.1 ± 0.6            | 14.6 ± 1.8                   |
> > > | Qwen2.5-32B   | 19.6 ± 0.6            | 32.1 ± 1.9                   |
> > >
> > > These results demonstrate that, while the `use code` tool is indeed utilized frequently, particularly on benchmarks involving arithmetic reasoning, it is **not the main driver** of the performance improvements. Rather, the full cognitive tools framework provides significantly better results.
> > >
> > > We will be glad to incorporate these quantitative results into the paper to substantiate our claims as we agree that they will give support to the robustness of our findings.

---

> > > > ### Comment · Reviewer_HC9H · 2025-08-07
> > > >
> > > > Thank you for the detailed rebuttal. I have no further concerns and would like to maintain my positive rating. I hope that all the revision experiments can be included in the final version of the paper.

---

### Note · Authors · 2025-08-12

We would like to thank the reviewers and the AC for their time, and the constructive feedback and suggestions. These encouraged us to conduct additional experiments and analyses that we believe have strengthened our paper. In particular, during the rebuttals we have provided:

* ablation studies on the effect of system prompt components (including the impact of motivational cues) and the role of code generation in our cognitive tools framework
* a detailed analysis of tool usage frequencies and interactions
* uncertainty intervals for the main accuracy results, along with significance tests to support our claims about the effectiveness of the cognitive tools framework compared to the standard baselines
* additional baselines, such as CoT and code-only setups, to better contextualize our results.

We are pleased that our additional experiments were helpful in addressing the raised concerns by the reviewers, and we are committed to incorporating the requested quantitative results and clarifications into the final version of the paper. We believe that these additions will improve the paper by providing a clearer context for our results and enhance the overall understanding of our findings, and are grateful to the reviewers for that.

---

### Decision · Program_Chairs · 2025-09-17

**Decision:**

Accept (poster)

**Comment:**

**summary**
This paper introduces a novel prompting framework that endows large language models with a set of "cognitive tools" inspired by cognitive psychology. This method operationalizes distinct reasoning steps, such as understanding the question, recalling related concepts, and backtracking, as explicit tools that the model can invoke within an agentic loop. The authors demonstrate that this approach significantly improves performance on challenging mathematical reasoning benchmarks for both open and closed-weight models.

**strengths**
- The core idea of modeling reasoning as a sequence of modular, cognitively-inspired tool calls is novel and compelling, offering a fresh perspective that bridges cognitive science with modern LLM agent frameworks.
- The empirical results are strong and consistent across a variety of models and challenging benchmarks. The performance lift on GPT-4.1 for the AIME2024 dataset is particularly impressive.
- The paper is well-written and logically structured, and the proposed framework allows for insightful analysis into the reasoning process itself, such as identifying patterns in tool usage based on task difficulty.
- The authors provided an exceptionally thorough rebuttal with extensive new experiments (e.g., CoT baselines, ablation studies) that effectively addressed the initial concerns raised by reviewers and strengthened the paper's claims.

**weaknesses**
- The evaluation is confined to the domain of mathematical reasoning. While this is a strong test case, the claims about eliciting general reasoning would be more robust if supported by experiments in other domains like planning or logical deduction.
- The performance improvement from using the full set of tools is often only marginally better than using the single best tool. A deeper analysis of the synergistic effects between tools would be beneficial to better understand their combined utility.


**final descision**
I recommend accepting this paper. It presents a novel, well-motivated, and empirically validated approach to enhancing LLM reasoning, and the authors' thorough rebuttal has convincingly addressed the initial critiques.